# All-natural 2D nanofluidics as highly-efficient osmotic energy generators

Jiadong Tang [1,3], Yun Wang[1,3], Hongyang Yang[2], Qianqian Zhang [1] ✉, Ce Wang[1], Leyuan Li[1], Zilong Zheng [1] ✉, Yuhong Jin[1], Hao Wang[1], Yifan Gu [2] ✉ & Tieyong Zuo[2]

Two-dimensional nanofluidics based on naturally abundant clay are good candidates for harvesting osmotic energy between the sea and river from the perspective of commercialization and environmental sustainability. However, clay-based nanofluidics outputting long-term considerable osmotic power remains extremely challenging to achieve due to the lack of surface charge and mechanical strength. Here, a two-dimensional all-natural nanofluidic (2D-NNF) is developed as a robust and highly efficient osmotic energy generator based on an interlocking configuration of stacked montmorillonite nanosheets (from natural clay) and their intercalated cellulose nanofibers (from natural wood). The generated nano-confined interlamellar channels with abundant surface and space negative charges facilitate selective and fast hopping transport of cations in the 2D-NNF. This contributes to an osmotic power output of ~8.61 W m$^{-2}$ by mixing artificial seawater and river water, higher than other reported state-of-the-art 2D nanofluidics. According to detailed life cycle assessments (LCA), the 2D-NNF demonstrates great advantages in resource consumption (1/14), greenhouse gas emissions (1/9), and production costs (1/13) compared with the mainstream 2D nanofluidics, promising good sustainability for large-scale and highly-efficient osmotic power generation.

As one of the most promising pollution-free renewable energy that generated by mixing seawater and freshwater with a salinity gradient, blue osmotic energy is considered as an ideal candidate to relieve energy and environmental concerns[1–3]. The membrane-based reverse electrodialysis (RED) technology is capable of directly capturing electrical power from osmotic energy based on a net ion flux stemming from ion concentration diffusion across an ion-selective membrane[4,5]. Accordingly, the development of high-performance ion-selective membranes is particularly critical to highly efficient osmotic energy harvesting[6–9]. In recent years, nanofluidics with nano-scale channels provided new opportunities for designing ideal ion-selective membranes, which can effectively regulate ion transport and contribute to

both strong ion selectivity and high ion flux, in consideration of the dense nanochannels and excessive surface charge[10–13]. Furthermore, ion rectification can be further achieved by controlling the shape and surface charge of nanochannels[14,15], which reduces ionic concentration polarization in a high-salinity and thus enhances osmotic energy harvesting[16,17].

The emerging nanofluidics present various configurations, among which the two-dimensional (2D) ones have shown great potential in large-scale osmotic energy utilization mainly because of their advantages in large-scale preparation and functionalization[18–20]. The 2D nanofluidics are known as the multi-layered membranes composed of self-assembly stacked 2D nanosheets, and the inter-layer space

[1]Key Laboratory of Advanced Functional Materials of Ministry of Education, College of Materials Science and Engineering, Beijing University of Technology, Beijing 100124, PR China. [2]Institute of Circular Economy, College of Materials Science and Engineering, Beijing University of Technology, Beijing 100124, PR China. [3]These authors contributed equally: Jiadong Tang, Yun Wang. ✉e-mail: zhangqianqian@bjut.edu.cn; zilong.zheng@bjut.edu.cn; guyifan@bjut.edu.cn

between the 2D nanosheets can be treated as lamellar nanochannels for selective ion transport[21]. In the past years, the potentials of some classical 2D materials in constructing nanofluidics were successively explored, such as graphene oxide (GO)[22], transition metal carbide and nitride (MXene)[23–25], boron nitride (BN)[26,27], and so on[3,28]. In spite of satisfactory osmotic power generation, the complex and expensive stripping or synthesis processes of 2D nanosheets, as well as massive use of ungreen reagents, restrict their real-world applications[29]. From the perspective of commercialization and environmental sustainability, the utilization of natural raw materials is one of the good alternatives to construct 2D nanofluidics toward the scaled blue osmotic energy harvesting and utilization.

Natural clay is an important carrier of 2D materials, featuring advantages of rich reserves, environmental friendliness, and good sustainability[30,31]. They have native layered crystal structures stabilized by weak van der Waals interaction[32,33], from which few-layer or even single-layer 2D nanosheets can be easily obtained via a simple ion-exchange exfoliation in mild water condition[34]. This provides a low-cost and eco-friendly route for developing 2D nanofluidics, compared with some mainstream 2D materials. It has been proved that some natural clay raw materials (such as vermiculite and montmorillonite (MMT)) could be readily developed into 2D nanofluidics for harvesting osmotic power[34,35], while the unsatisfactory mechanical strength usually limits their durability and thus affect energy output performance[36,37]. Nanofiber intercalation is known as an effective way to improve the mechanical strength of 2D nanofluidic membranes. Some robust 2D nanofluidics (GO, BN, and MoS$_2$) were recently constructed based on the layered 2D nanosheets with interlamellar nanofibers, such as aramid nanofiber (ANF)[38–40], cellulose nanofiber (CNF)[3,41,42] and silk nanofiber[43,44]. Among them, the CNF is a kind of natural nanofiber derived from plant cellulose fibers that have a great economic advantage (only ~20 USD kg$^{-1}$) for industrial-scale manufacturing[45]. When serving as intercalators, the intrinsic high mechanical strength of CNFs could first strengthen the 2D nanofluidic membranes. More importantly, the formation of oxo-bridging between dense oxygen-containing groups on the CNF molecules and 2D nanosheets could further stabilize the membrane with interlocking configuration[46,47]. Therefore, it is desired to develop robust all-natural 2D nanofluidics based on the clay base and the CNF intercalator for achieving sustainable osmotic power output.

Herein, we demonstrate the use of all-natural materials to construct 2D nanofluidics for high-efficiency and large-scale blue osmotic energy harvesting. The 2D natural nanofluidics (2D-NNF) are based on a robust and flexible membrane with interlocking configuration between MMT matrix and the intercalated CNFs. The interlayer nanochannels cooperated with the negatively-charged surface contribute to a high cation selectivity, which is the basis of osmotic energy harvesting. Furthermore, the membrane thickness could be controlled as low as 5 μm, benefiting for a low internal resistance and thus high osmotic power output. Consequently, the 2D-NNF delivers a considerable power density of 8.61 W m$^{-2}$ in the simulated sea/river concentration gradient (50-fold) condition, which is superior to all state-of-the-art 2D nanofluidics (~5 W m$^{-2}$). For the large-scale membrane with an area of 700 cm$^2$, the osmotic energy conversion is carried out by selecting several test sites from the different regions of membrane under the same test conditions. The average maximum power of different test sites still reaches 8.36 W m$^{-2}$ and could maintain long-term stability over 30 days, attributing to excellent uniformity and stability in both physical and chemical structures. Beyond that, detailed technoeconomic analysis and life cycle assessment (LCA) reveal that the 2D-NNFs possess predominant economic, energy, and environmental benefits compared with their classical analogs such as GO and MXene-based nanofluidics, which further highlights the advantages of sustainable and all-natural system developed in this work as the large-scale blue osmotic energy harvesting platform.

## Results

The raw materials for constructing all-natural 2D nanofluidics (2D-NNF) were all derived from the natural elements, i.e., clay and tree as illustrated in Fig. 1a, in which the negatively-charged MMT and nanofibers both contributed to a selective and fast transport of cations, which is the key to achieve high-performance osmotic energy harvesting. The natural bulk bentonite was firstly exfoliated to 2D MMT nanosheets (Fig. 1b, left), whose size was ~300 nm and the thickness was ~1.60 nm (Supplementary Fig. 1a), as previously reported[29,36]. The as-prepared MMT nanosheet exhibited negative charge because of the lower state atomic substitution in the lattice[48], which had a negative zeta potential of −27.5 mV (Supplementary Fig. 2a). The raw wood was separated into CNFs via a pulping-oxidation-dispersion process, which had a micrometer scale length and an average diameter of ~6.76 nm (Fig. 1b, right and Supplementary Fig. 1b). Such a CNF was negatively charged owing to plenty of carboxyl and hydroxyl groups on the cellulose molecules, which was confirmed by the negative zeta potential (−34.2 mV) (Supplementary Fig. 2a). Then, the obtained MMT nanosheets and CNFs were mixed evenly and dispersed well in the deionized water (Supplementary Fig. 2b), followed by a vacuum filtration to form the all-natural 2D nanofluidics. The as-prepared 2D nanofluidics appeared as a paper-like self-standing membrane with orderly stacked lamellar configuration (Fig. 1c), in which MMT nanosheets were the base and CNFs as the intercalators. The membrane thickness was as low as several micrometers, contributing to a low internal resistance for efficient osmotic energy harvesting. Meanwhile, an interlocking configuration was formed based on the chemical bonding between MMT nanosheets and CNFs, which was confirmed by the X-ray photoelectron spectroscopy (XPS) and Fourier transform infrared (FTIR) spectroscopy characterizations. The fine spectra of Al 2p and Si 2p demonstrated obvious chemical shifts for the characteristic peaks of 2D-NNF compared with pristine MMT (Supplementary Fig. 3), attributing to the formation of covalent bonds (Si−O−C and Al−O−C) between MMT and CNFs[49,50]. What's more, a characteristic peak appeared in 848 cm$^{-1}$ of the FTIR spectra could be assigned to the Al−O−C vibrations[49], indicating the formation of covalent binding between MMT nanosheets and CNFs (Supplementary Fig. 4). Consequently, the chemical binding between MMT nanosheets and CNFs contributes to an interlocking configuration formed in the 2D-NNF, which is similar as the stable "brick and mud" configuration of natural nacre[43]. Such a configuration stabilized the 2D nanofluidics and thus enabled robust large-scale membrane with a diameter of ~30 cm (Fig. 1d). Compared with pristine MMT membrane, the 2D-NNF with 10% CNF content demonstrates a high tensile strength up to ~115 MPa (Fig. 1e), which is comparable with natural nacre (~130 MPa)[38,51]. The Young's modulus also achieved a considerable value of about 3.60 GPa (Supplementary Fig. 5). Furthermore, the as-prepared membrane had good hydrophilicity with the contact angle of about 45° (Supplementary Fig. 6), providing the necessity for effective osmotic energy harvesting.

The nanoconfined channels comparable to electric double layer thickness (EDL) were one basic to achieve good ion selectivity. The pristine MMT nanofluidics had an interlamellar nanochannel size of 1.22 nm obtained from the small-angle X-ray diffraction pattern, which increased with the content of intercalated CNFs (Fig. 1f). Notably, the interlayer wide of the 2D-NNF was additionally expanded about 0.61 nm in fully hydrated state (Supplementary Fig. 7 and Supplementary Note 1), but the interlayer spacing was still less than the nanofibers diameter. This was attributed to that a few nanofibers were randomly intercalated into the lamellar membrane in the preparation of 2D-NNF, leading to a random expansion in some spaces of the nanochannels[52] (Supplementary Fig. 8). While, the interlayer wide obtained from the SAXD was a feedback to the periodic parallel arrangement of MMT lamellas, and the nonperiodic CNF intercalated regions cannot be reflected from this measurement. Therefore, the

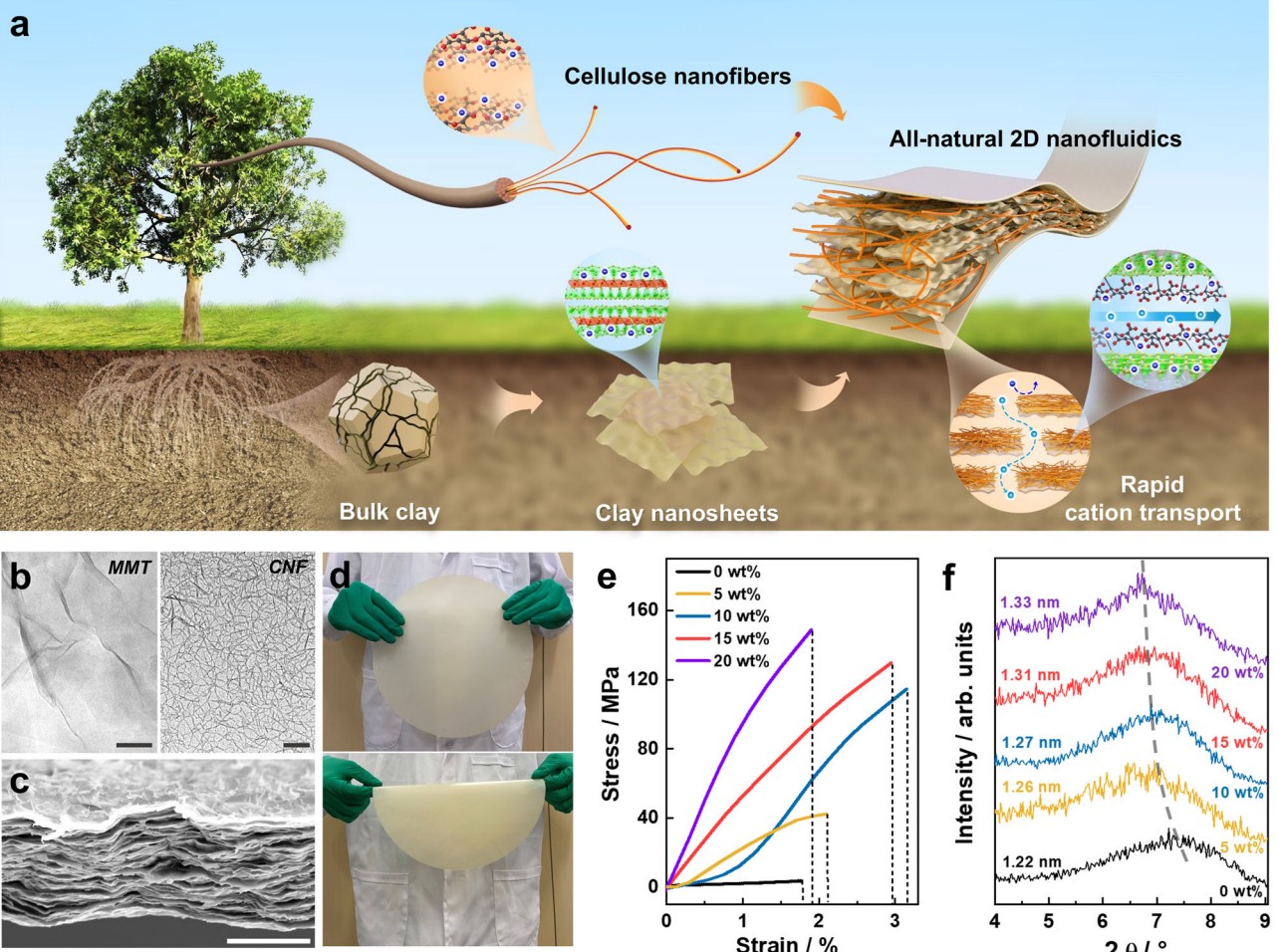

**Fig. 1 | Characterization of 2D natural nanofluidics (2D-NNF). a** Schematic diagram for assembly of natural montmorillonite (MMT) and natural cellulose nanofibers (CNFs) into a large-area 2D-NNF membrane with selective and fast cation transport nanochannels. The inset picture of CNF (left): C, dark; O, red; H, white; the inset picture of clay nanosheets (left): Si, green; O, yellow; Al, red; and the dark blue spheres (left) indicated negatively charged, while the light blue spheres (right) indicated cations. **b** TEM images of the as-prepared MMT nanosheets (left) and CNFs (right). Scale bars: 200 nm. **c** Cross-sectional SEM image of the 2D-NNF with CNF content of 10%, showing an ordered lamellar structure. Scale bar: 5 μm. **d** Photographs of flat state (upper) and bent state (bottom) of a robust large-scale 2D-NNF membrane (φ 30 cm). **e** Tensile stress-strain curves of the 2D-NNF with different CNF contents. **f** SAXRD patterns of the 2D-NNF with various CNF contents. The dotted line indicates the layer spacing (left) gradually increasing with the content (right) of intercalated CNFs. Source data are provided as a Source Data file.

measured interlayer spacing (sub-2 nm) was smaller than the average diameter (~6 nm) of nanofibers. Another basic to construct high-performance cation-selective nanofluidics is excess negative charge. The excess negative charge on MMT wall is termed as surface charge and the negative charge from the intercalated CNFs is usually called space charge[3,38]. The space charge of CNFs could enlarge the EDL region in the nanochannels, contributing to higher cation selectivity and faster cation transport[38,53,54] (Supplementary Fig. 8). According to the zeta potential results, the 2D-NNF carried negative charges at a wide ion concentration range (0.1 mM–1000 mM, Supplementary Fig. 9a) and increased with solution pH (Supplementary Fig. 9c). While the surface charge density of 2D-NNF is highly dependent on the ion concentration of solution (Supplementary Fig. 9b). In this case, modulation of solution pH will inevitably introduce excess $H^+$ or $OH^-$ ions and thus increase the ion concentration of solution, especially for low-concentration and extreme pH values. Therefore, the surface charge density evolution with varied solution pH was calculated considering the concentration increase of extra $H^+$ or $OH^-$ ions (Supplementary Note 2). The results show that the charge density is higher at the two extreme pH (3, 11) because a higher ion concentration shortens the Debye length ($\lambda_D$) value (Supplementary Fig. 9d and Supplementary Table 1). For the medium pH value (5, 7, 9), the concentration variation

caused by few extra ions is almost negligible, and thus the charge density increases with pH value as that of zeta potential. Therefore, the 2D-NNF with high surface charges and excellent mechanical strength are expected to be an ideal candidate for highly efficient osmotic power generation.

## Transmembrane ionic transport properties

The transmembrane ionic transport properties of the 2D-NNF were investigated by experimental design and theoretical simulations. Here, KCl was selected as the representative electrolyte because of the similar ion mobility of cation ($K^+$, $1.960 \times 10^{-9}$ m²s⁻¹) as that of anion ($Cl^-$, $2.032 \times 10^{-9}$ m²s⁻¹). The optimized 2D-NNF of 10%-CNF content and 5.3 μm thickness was used for the following measurements, which were determined by the osmotic energy conversion behavior (Supplementary Fig. 10). Firstly, a series of current-voltage (*I-V*) curves were recorded in KCl electrolyte with concentration from 0.01 to 1 M (Supplementary Fig. 11a). All of them presented the linear ohmic behavior, from which the transmembrane ionic conductance could be calculated. As shown in Fig. 2a, the ionic conductance follows the bulk rule with a linear relationship with KCl concentration in the high-concentration region (blue). When the concentration reduced to below 10 mM, the ionic conductance gradually deviated from the bulk

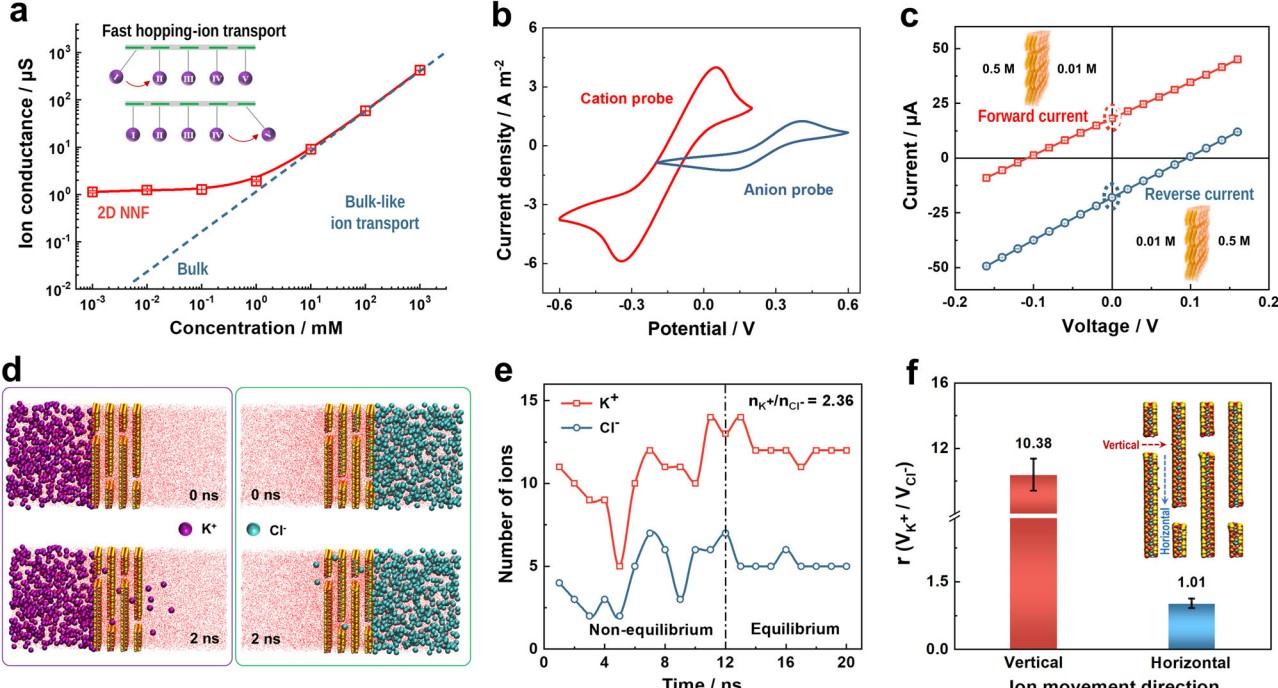

**Fig. 2 | Transmembrane ionic transport properties. a** Ionic conductance of 2D natural nanofluidics (2D-NNF) as a function of electrolyte (KCl) concentration. The ionic conductance deviates from bulk value (red line) at low concentration region, indicating the surface-charge-controlled ionic transport behavior. The inset pictures illustrate rapid ion transport through a hopping mechanism, where I, II, III, IV, and V represent the cations ($K^+$) adsorbed by the electric double layer. The error bars in the figure represent the standard deviations for three measurements. **b** CV curves of the ITO-supported 2D-NNF using $[Ru(NH_3)_6]^{3+}$ as a cationic electroactive probe (red line) and $[Fe(CN)_6]^{3-}$ as an anionic electroactive probe (blue line). **c** *I-V* curves of 2D-NNF recorded in 50-fold KCl concentration gradient ($C_{high}$ = 0.5 M, $C_{low}$ = 0.01 M) under forward and reverse diffusion directions. The error bars in the figure represent the standard deviations for three measurements. **d** MD simulation snapshot of ion permeation system after equilibrium state 0 ns and 2 ns in 2D-NNF nanochannel. The red area in the middle of the picture represents water molecules. **e** The number of ions passing through nanochannels with time in the MD simulations. The equilibrium structure was achieved at 12 ns of MD simulations. The vertical dashed lines in the figure represent the dividing lines for distinguishing between equilibrium and non-equilibrium regions. **f** Evolution of the calculated transport rate ratios between vertical and horizontal direction (inset) $K^+$ ions and $Cl^-$ ions under a transmembrane electric field of 0.5 V $nm^{-1}$. The error bars in the figure represent the standard deviations for six statistics. Source data are provided as a Source Data file.

value (red line) and turned to a plateau at low concentrations (<$10^{-1}$ mM), which could be ascribed to the surface-charge-governed ion transportation led by the EDL[38,44]. In aqueous solution, the negatively-charge surface of 2D-NNF could adsorb counterions to form EDL in the nanochannels. The EDL region plays a leading role in regulating ion transport to achieve cation selectivity and fast cation conduction[55-57]. Consequently, the nanochannel with full-filled EDL demonstrates superior ion regulation behaviors and thus a high osmotic energy harvesting. The thickness of the EDL was inversely proportional to the electrolyte concentration, that is the low concentration with the large thickness of the EDL, as previously reported[58,59]. And the expansion behavior of 2D-NNF nanochannels depended on the applied solutions (KCl) with different EDLs was further verified (Supplementary Fig. 12). In the low concentration region (pink), the EDL thickness (Debye length, $\lambda_D$) was closed to or larger than the channel radius (Supplementary Table 1), resulting in an overlapping EDL and a surface charge-governed ion transport behavior. Here, the ion transport is believed to follow the hopping mechanism (Fig. 2a, inset)[60]. To understand the hopping mechanism of cation diffusion process, the simulations of $K^+$ ion transport in negatively-charged 2D nanofluidic membrane were performed via Ab initio molecular dynamics (MD). Diffusion EDL was mainly derived from Gouy−Chapman−Stern model[61], which consists of the Stern layer and diffuse layer. While the diffuse layer was composed of $K^+$ in our modeling 2D-NNF/$K^+$ system. The 2D-NNF was coated with a layer of $K^+$, and each $K^+$ has interactions with the O at the surface of 2D-NNF. At the beginning of MD simulations, the additional $K^+$ with velocity collided with one $K^+$ at rest in the diffuse layer, and the other $K^+$ in the diffuse

layer moved with pendulum-like motion, along the additional $K^+$ velocity direction. $K^+$ far away from the collision moved 1.5 nm following 80 ps in the diffuse layer, through pendulum-like motion shown in Supplementary Fig. 13. Thus, the rapid $K^+$ transport in developed 2D nanofluidics presented energy transfer hopping process according to the hopping mechanism.

Then, cyclic voltammograms behaviors of the 2D-NNF in the presence of electroactive redox probes, $[Ru(NH_3)_6]^{3+}$ and $[Fe(CN)_6]^{3-}$ were studied to demonstrate the cation selectivity (Fig. 2b). Both of these species have characteristic oxidation and reduction peaks upon potential scanning[62]. While the 2D-NNF exhibited a strong and highly selective electrochemical response for the transport of the cationic probe $[Ru(NH_3)_6]^{3+}$ (red trace), and a weak response for the anionic probe $[Fe(CN)_6]^{3-}$ (blue trace). The reaction charges are ~8.44 C $m^{-2}$ and 2.52 C $m^{-2}$, respectively (Supplementary Fig. 11b and Supplementary Note 3), corresponding redox peak area when using $[Ru(NH_3)_6]^{3+}$ and $[Fe(CN)_6]^{3-}$ probes. The marked contrast results from the preferential cation diffusion. Figure 2c shows the forward and reverse diffusion *I-V* curves of the 2D-NNF measured under a 50-fold concentration gradient (0.5 M/0.01 M KCl). Under a reverse concentration gradient, the open-circuit voltage ($V_{OC}$) and short-circuit current ($I_{SC}$) had the similar values, but with the different polarity due to the opposite ion diffusion, and it demonstrated that the 2D-NNF had a symmetric structure without preferential direction for ion diffusion[38]. Furthermore, the ion selectivity can be further measured by an *I-V* test with various KCl concentration gradients over the membrane. Saturated KCl salt bridges were used to eliminate the redox potential contribution from the unequal potential

drop at the electrode-solution interface. As the 2D-NNF is cation-selective, it can transport cation (i.e., K$^+$) preferentially from high concentration side to low concentration side, generating the osmotic current ($I_{OS}$) and osmotic potential ($V_{OS}$)[63]. By collecting $I$-$V$ curves in the presence of different transmembrane concentrations, the values of $V_{OS}$ and $I_{OS}$ can be directly obtained (Supplementary Fig. 14a and Supplementary Table 3). Therefore, the ion transfer number ($t^+$) that quantifies the ion selectivity can be calculated (Supplementary Note 5). Under a 10-fold (1 mM/10 mM KCl) concentration gradient, the transference number can reach up to about 0.92 (Supplementary Fig. 14b). The CNF intercalation would increase the space charges in the MMT nanochannels and thus achieve a considerable cation selectivity.

In order to understand the mechanism of the ion permeation in 2D-NNF channels, we investigated the migration of both K$^+$ and Cl$^-$ ions in 2D-NNF channels employing MD simulations. The modeling system of $5.2 \times 13.5 \times 30.0$ nm$^3$ was built (Supplementary Fig. 15), which has a total of 197975 atoms for 2D-NNF/KCl/H$_2$O configuration, and the whole system is electrically neutral. While, there are 1086 K$^+$ cations and 1086 Cl$^-$ anions, which are composed of 1 mol/L KCl solution with 58161 water molecules. The modeling 2D-NNF was constructed with the layer distance of 1.90 nm (Supplementary Fig. 7b), which was obtained from XRD patterns. The negative charges surface were induced by adjusting the Mg:Al atomic ratio in the 2D-NNF structure (Supplementary Note 7). The surface charges of 57.37 mC m$^{-2}$ were obtained, which were derived from the ratio of the total charges of ions on the 2D-NNF surface in MD equilibrium structure, and it was consistent with the measured 65.64 mC m$^{-2}$ in experiment (Supplementary Fig. 9b). In MD simulations, the equilibrium structure with 2D-NNF width of 7.6 nm was obtained, following at 12 ns MD simulations, in order to investigate the ion selectivity between two electrolyte layers (Supplementary Fig. 16). The snapshots of K$^+$ and Cl$^-$ ion permeation were presented following the equilibrium structure for 2 ns, as shown in Fig. 2d. In the beginning, the location of K$^+$ (or Cl$^-$) ions were restricted in one side of 2D-NNF layers, and then the ions permeated through the 2D-NNF via the pore channels to the other side. The ions diffusing numbers through the 2D-NNF layers were counted, starting from MD equilibrium structure. As the permeated ion numbers increased, the increment of K$^+$ ions was much higher than that of Cl$^-$ ions (Fig. 2e). When the equilibrium structure was achieved following 12 ns MD simulations, the diffused ion numbers through 2D-NNF hardly changed anymore, and the migration ratio between K$^+$ and Cl$^-$ ions in 2D-NNF was exported, which is 2.36 (Fig. 2e).

Moreover, the ionic transport velocity in nanochannels was further investigated. The ion transport in 2D lamellar could be divided into vertical and horizontal direction relative to 2D-NNF growth orientation (Fig. 2f, inset). The velocity ratio ($V_K^+/V_{Cl}^-$) between K$^+$ and Cl$^-$ ions in the horizontal direction was 1/10 of that $V_K^+/V_{Cl}^-$ in vertical direction when ionic transport achieved the equilibrium in MD simulations (Fig. 2f, Supplementary Fig. 17 and Supplementary Table 4). The difference in the transport velocity between vertical and horizontal directions was observed for bulk-like ion transport (1 M KCl), which was attributed to the electrostatic and dehydration of ions when entering the vertical channel. Firstly, the negative surface drove up the K$^+$ vertical velocity, considering the attractive electrostatic force in vertical forward direction. However, that was opposite case for anion (Cl$^-$) transport, the negative surface lowered down the Cl$^-$ vertical velocity, considering repulsive electrostatic force. Therefore, the permeating velocity of K$^+$ cation was around twice as large as that of Cl$^-$ anion (Supplementary Fig. 18). Meanwhile, DFT calculations implied the hydration energy of K$^+$ is -3.07 eV, while that of Cl$^-$ is about 3.72 eV (Supplementary Fig. 19). Consequently, K$^+$ undergoes the dehydration process more easily than Cl$^-$ when entering the channel. Once inside the horizontal channel, the ions complete the dehydration process, and the electrostatic forces on both sides of the nanochannel are

symmetric. Therefore, MD simulations showed similar velocities for both K$^+$ and Cl$^-$ ions along the horizontal direction. In the future, the fast cations transport rate can be achieved by increasing the pore density on 2D-NNF and decreasing the horizontal transport process.

## Osmotic energy generation performance

The high surface charge and excellent ion-transmembrane-transport ability of the 2D-NNF paved the way for salinity gradient power harvesting. As illustrated in Fig. 3a, the osmotic energy existing in natural seawater and river water can be harvested through the 2D-NNF with the help of electrochemical redox reactions on the electrode surface. The $V_{OC}$ and $I_{SC}$ can be obtained from different salt concentration gradients (Supplementary Fig. 21a). The $V_{OC}$ was increased from 73 to 167 mV at pH ~5.7 (NaCl) with varying the gradients from 10-fold to 5000-fold. Moreover, the 2D-NNF operated over 2000 s under various concentration gradients (Supplementary Fig. 21b) and exhibited excellent current output stability. Under a 50-fold transmembrane concentration gradient (0.5 M/0.01 M NaCl), $V_{OC}$ and $I_{SC}$ were observed on the coordinate axes of the measured $I$-$V$ curve, to give values of 17.85 μA and −108 mV, respectively (Fig. 3b). The diffusion current ($I_{diff}$) and diffusion potential ($E_{diff}$) can be further corrected by subtracting the contribution of the redox potential of the electrode (Fig. 3b inset and Supplementary Note 4). The harvested power could be further output to an external circuit to supply a load resistance ($R_L$). In an artificial seawater and river water salinity gradient (0.5 M/0.01 M NaCl), the energy conversion performance of the 2D-NNF was investigated. According to the generated current ($I$), the output power density ($P$) can be calculated with the equation, $P = I^2 \times R_L$. As shown in Fig. 3c, the diffusion current density gradually decreased with the increase of load resistance, while the output power density achieved a maximum value of ~8.61 W m$^{-2}$ with a low resistance of ~6 kΩ. The corresponding energy conversion efficiency reached about 21% (Supplementary Note 6). Meanwhile, the current density and power density under different concentration gradients were investigated by gradually increasing the concentration from 1 mM to 100 mM at the low-salinity side. Considering that the salinity concentration of seawater, the high-salinity side was set as 0.5 M (NaCl). Along with the concentration gradient increasing, the produced current density gradually increased, and the output power density reached up to 15.87 W m$^{-2}$ at 500-fold salinity gradient (Fig. 3d and Supplementary Fig. 22), demonstrating the 2D-NNF could be adapted to different river water concentration conditions. Meanwhile, the measured $V_{OC}$ and $I_{SC}$ values were consistent with the power output variation (Supplementary Fig. 23). Furthermore, a maximum power density of 60.46 W m$^{-2}$ could be achieved (Supplementary Fig. 24) from the 2D-NNF-based osmotic energy harvesting system at the salt lake/river condition (4.3 M/0.01 M NaCl).

For a fair comparison, the same salinity gradients were applied to the KCl aqueous solution system, that is the high-salinity solution was fixed at 0.5 M KCl and the low-salinity one was controlled at 1, 10, and 100 mM. In this case, the obtained osmotic power density was, respectively 2.82 (5-fold), 10.38 (50-fold), and 17.06 W m$^{-2}$ (500-fold) as shown in Supplementary Fig. 25. A higher osmotic energy output in KCl system compared with NaCl system was mainly attributed to the higher ionic diffusion coefficient of K$^+$ ion ($1.960 \times 10^{-9}$ m$^2$ s$^{-1}$) than Na$^+$ ion ($1.334 \times 10^{-9}$ m$^2$ s$^{-1}$). Notably, when $R_L$ was connected to extract the generated power, the maximum extractable power reached its maximum when $R_L = R_M$ (membrane internal resistance)[62]. The lower the membrane resistance, the more efficient the ion transport will occur, which was favorable to efficient osmotic energy conversion. Therefore, the membrane resistance in different salinity gradients was studied by diffusion voltage and diffusion current. For the 2D-NNF (5 μm), the measured $R_M$ decreased from 583 kΩ to 30 kΩ as the concentration gradient increased from 10 to 5000, and the $R_M$ increased as the thickness increased, which can be attributed to the increase in ion

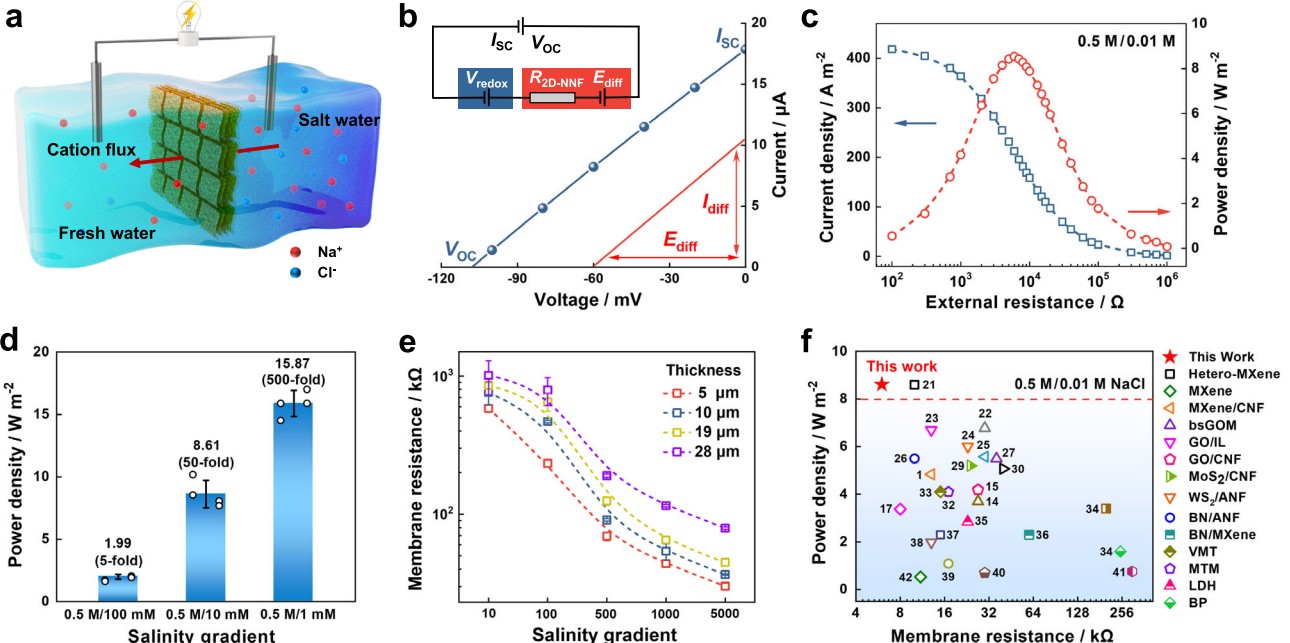

**Fig. 3 | Osmotic energy generation performance. a** Schematic of the osmotic energy conversion based on the 2D-NNF. **b** *I-V* curves of the membrane under a transmembrane salinity gradient (0.5 M/0.01 M NaCl) before (blue line) and after (red line) the subtraction of the contribution of redox potential. The inset picture shows the equivalent circuit of the power source. **c** The output power density and current density as the functions of load resistances. Power output under 0.5 M/ 0.01 M NaCl salinity gradients, which exhibits a maximum value of -8.61 W m$^{-2}$ at the load resistance of -6 kΩ. **d** The output power densities at different concentration folds. The high-salinity solution is fixed at 0.5 M NaCl, and low-salinity is varied from 1 mM to 100 mM. The maximum value is 15.87 W m$^{-2}$. **e** Membrane resistances at different thicknesses under a series of concentration gradient. The low concentration side is set to 0.1 mM. **f** Comparisons of osmotic energy harvesting capacity of 2D-NNF with the reported 2D nanofluidics, the reference numbers are from Supplementary Table 5. The error bars in (**d**) and (**e**) represent the standard deviations for four measurements. Source data are provided as a Source Data file.

transport paths across the membrane and the decrease in ion flux (Fig. 3e). To the best of our knowledge, the 2D-NNF has a superior osmotic energy harvesting behavior (power density of 8.61 W m$^{-2}$ at a lower resistance of 6 kΩ) compared with all previously reported 2D nanofluidics (Fig. 3f). However, the power density of 2D-NNF was lower than some state-of-the-art 1D and 3D nanofluidics (Supplementary Table 5) due to the generally accepted ion transport paths differences caused by distinct nanochannel configurations, which provides the further research directions for optimum structural design of 2D-NNF.

## Ion-dependent osmotic energy generation

In order to investigate how the ionic electrostatic interaction, diffusion coefficient, and hydration diameter affected the osmotic energy conversion behavior of 2D-NNF, various salt solution electrolyte systems were studied under the same concentration gradient (0.01 M/0.5 M). For the monovalent ions, the membrane-based RED system could deliver the maximum power output for the KCl (~10.38 W m$^{-2}$) and minimum power output for the LiCl (~5.97 W m$^{-2}$) (Fig. 4a, b), which is attributed to the different diffusion coefficients of ions (K$^+$ > Na$^+$ > Li$^+$)[64]. The faster the cation diffuses with equal valent, the more efficient the charge separation will occur[6,65], which also corresponds to the cation selectivity characteristics of the 2D-NNF verified by experimental and theoretical simulations. For the divalent cations, on the one hand, the diffusion coefficients of Ca$^{2+}$ (0.793 × 10$^{-9}$ m$^2$ s$^{-1}$) and Mg$^{2+}$ (0.705 × 10$^{-9}$ m$^2$ s$^{-1}$) are lower than that of Li$^+$ (1.029 × 10$^{-9}$ m$^2$ s$^{-1}$) caused a lower power density; on the other hand, stronger electrostatic interaction due to their large intrinsic charge is also an important factor hindering their power output[6]. As shown in Fig.4c, MD simulations presented that the binding energy between divalent cations (~15 kJ mol$^{-1}$) and nanosheets was much higher than those of monovalent cations (~11 kJ mol$^{-1}$). While it confirmed there are stronger electrostatic interactions existing for divalent cations rather than monovalent cations. Moreover, the large hydration

diameter of divalent cations further limits the migrations in nano-confinement space, and results in the low migration velocity ratio ($V_{cations}/V_{Cl^-}$) and diffusion coefficient (Fig. 4d). Therefore, divalent cations (Ca$^{2+}$ and Mg$^{2+}$) had lower output power density than those monovalent cations (Fig. 4b). It is worth noting that the power output of the divalent Mg$^{2+}$ ion was slightly higher than that of the Ca$^{2+}$, which could be attributed to the relatively low binding energy between the Mg$^{2+}$ ion and the nanosheet. Moreover, the energy conversion of 2D-NNF with different pH conditions was also tested under a 50-fold salinity concentration (0.5 M /0.01 M KCl). It can be seen from the current and power output curves increase with the increase of pH (Supplementary Fig. 26a), which could be attributed to the deprotonation of carboxyl groups on the surface of CNF. As a result, the corresponding output power density reached the maximum (-13.80 W m$^{-2}$) at pH = 11 (Supplementary Fig. 26b). Furthermore, the output current, power density, and energy conversion efficiency of 2D-NNF gradually increased with the temperature (Supplementary Fig. 27) due to a high temperature will reduce the liquid viscosity and increase the ionic mobility[66,67], and the average output power could maintain at about 14.43 W m$^{-2}$ for a long-term measurement at 313 K. These results implied the potential for harvesting osmotic power from industrial wastewater, which may have a wide range of temperatures and pH values, from strongly acidic to strongly basic.

## Large-scale 2D-NNF generating osmotic energy toward practical applications

The osmotic energy conversion performance was studied based on a large-area 2D-NNF with a diameter of 30 cm toward the real-world applications. To evaluate the reversibility of the scaled-up membrane, three large-area 2D-NNF were prepared using the same process, and then nine different regions were selected from each membrane for the thickness and interlayer spacing characterizations (Supplementary

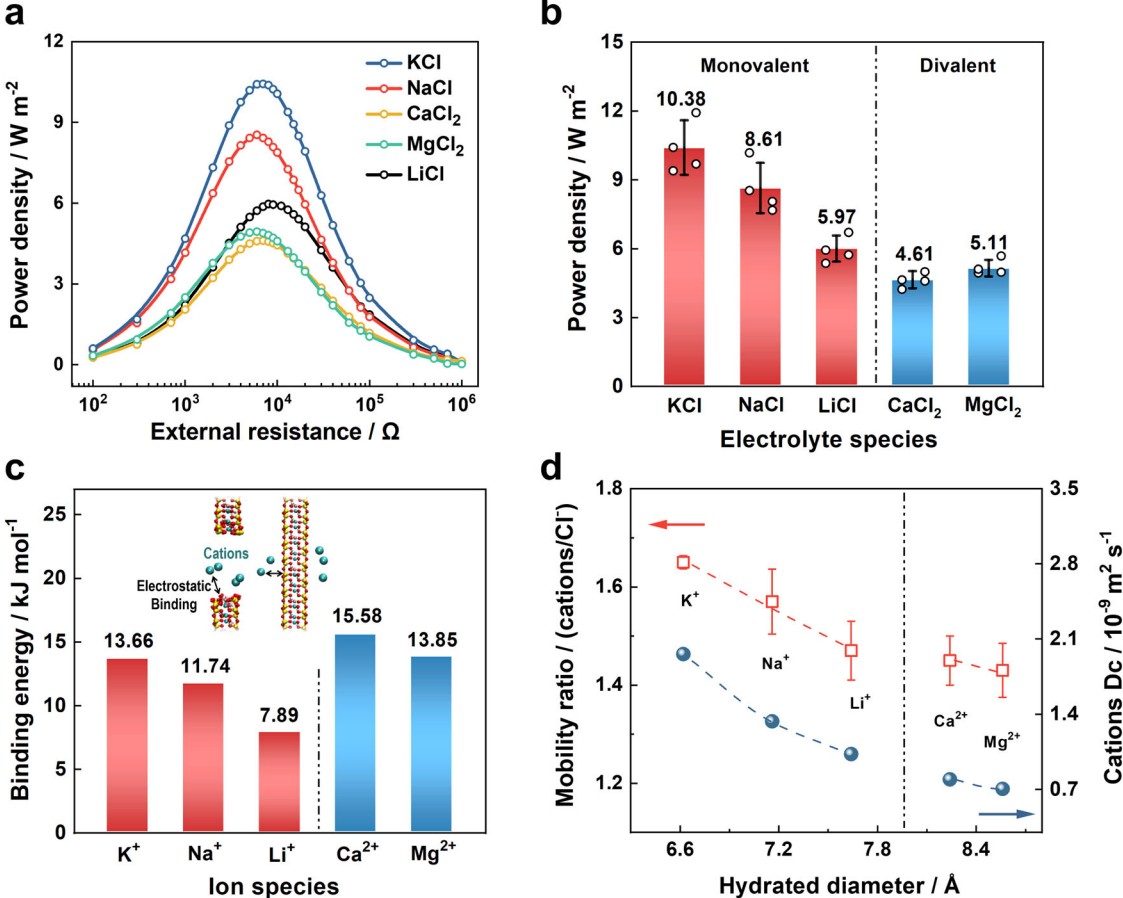

**Fig. 4 | Ion-dependent osmotic energy generation. a** The power densities at different electrolytes as a function of the external resistance. **b** The maximum power generation under different species of ion at a 50-fold salinity gradient (0.5 M/0.01 M). **c** The calculated binding energy between monovalent or divalent cations and nanosheets. Insets show the configurations of the binding energy between cations and unit cell. **d** Ion mobility ratio and cation diffusion coefficient relating to the hydrated cation diameter calculated by MD simulation under 1 M electrolyte system. The error bars in (**b**) and (**d**) represent the standard deviations for four measurements. All the vertical dashed lines in the figure represent the dividing lines for distinguishing between monovalent and divalent cations. Source data are provided as a Source Data file.

Fig. 28). It was found that all the membranes had almost the same thicknesses (~16 μm, Supplementary Fig. 29) and interlamellar spacing (1.24 nm, Supplementary Fig. 30), indicating good consistency and reversibility for the scale-up of the 2D-NNF. Furthermore, the thickness of the large-area membrane could be flexibly regulated by controlling the preparation process (Supplementary Figs. 31 and 32). Consequently, four sites were randomly selected on the 2D-NNF for the following osmotic energy conversion studies (Fig. 5a). It was found that the output curves of current density and power density at the four randomly selected test sites kept a good agreement (Fig. 5b) by mixing seawater and river water (0.5 M/0.01 M NaCl). The measured power density of the amplifying 2D-NNF at the four sites reached about 8.24 W m⁻², 8.04 W m⁻², 8.33 W m⁻², 8.36 W m⁻², respectively (Fig. 5c), which indicated it still remained a good osmotic energy conversion performance after amplification, in consideration of the homogeneous dispersion of precursor solution (Supplementary Fig. 33) and the stability of natural materials. In the future, large-scale fabrication can be achieved by designing a continuous pressure-filtration system. This strategy may provide a renewed understanding of the potential of 2D material-based nanofluidics, adding a prospect for future industrial applications. Furthermore, the output current density could maintain 95.9% of the maximum value after a continuous operation for 160 min (Fig. 5d), because the 2D-NNF was capable of stabilizing local concentration gradient and thus promoting continuous power generation. Meanwhile, the 2D-NNF exhibited admirable stability and high

endurance. The output power density did not show obvious attenuation after 30 days (Fig. 5e and Supplementary Fig. 34). Besides, the 2D-NNF were still robust enough and maintained chemical stability even after long-term immersing in water (Supplementary Fig. 35), and the nanochannel space also still remained stable in both dry and hydrated conditions (Fig. 5e and Supplementary Fig. 36), which was ascribed to the introduction of natural CNFs increased the stability of nanofluidics. Furthermore, the output current was significantly enhanced compared with small-area membrane (several microamperes) when the permeability area was enlarged to 314 cm² (Supplementary Fig. 37), and remained stable for a long term over 120 min (Supplementary Fig. 38). The results show that the 30 cm-diameter membrane could be directly used for osmotic energy harvesting and output considerable current and voltage, indicating great potentials in real-world applications. In a word, the 2D-NNF maintained high mechanical strength and high ion selectivity while possessed an extremely low resistance (~6 kΩ) advantage by comparison with the mainstream 2D material nanofiber-based nanofluidics (Fig. 5f and Supplementary Table 6), resulting in the highest power density output. Therefore, the 2D-NNF is expected to be a candidate for high-efficiency osmotic energy conversion in the future.

## Resource, environmental, and technoeconomic analysis

Detailed production processes of 2D materials and nano cellulose used for nanofluidics preparations in reported works[29,38,39,41,42] were

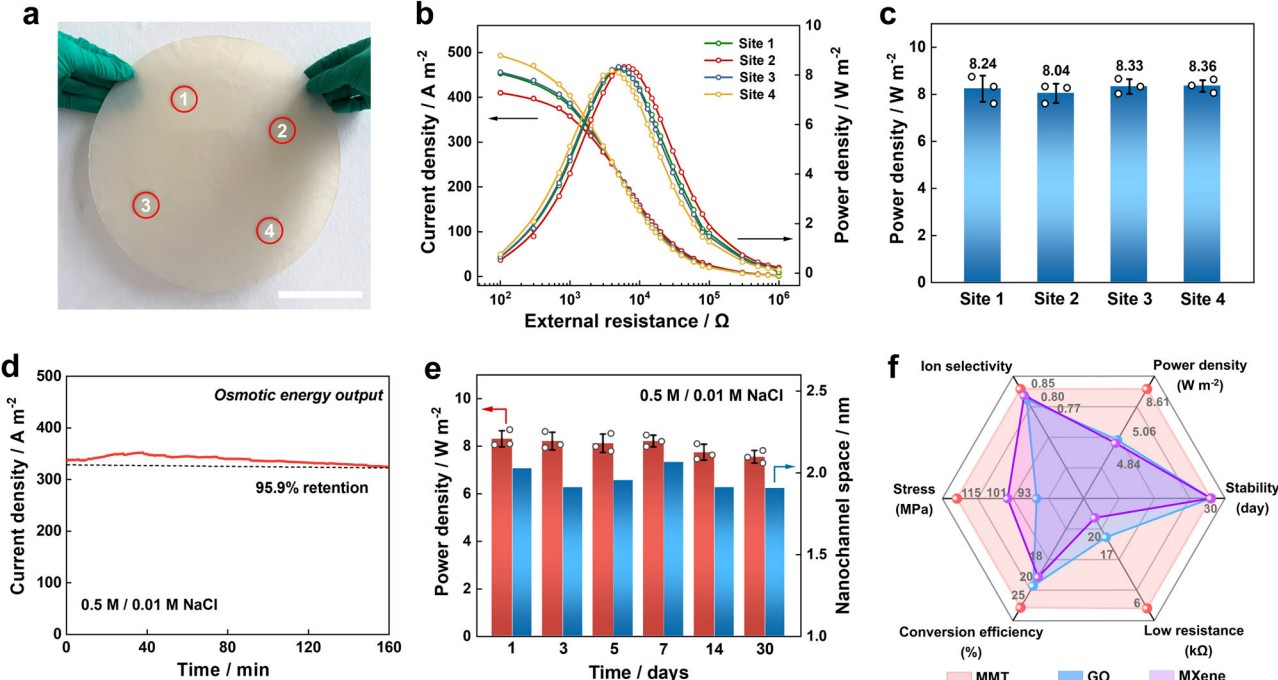

**Fig. 5 | Large-scale 2D-NNF generating osmotic energy toward practical applications. a** The photograph of large-scale (>700 cm$^{-2}$) free-standing 2D-NNF marked with four randomly selected test sites. Scale bar: 10 cm. **b** Current density and power density evolutions with the external resistance at four selected sites by mixing artificial seawater (0.5 M NaCl) and river water (0.01 M NaCl). **c** Maximum power generation at different sites of the large-scale 2D-NNF. **d** Lifetime of osmotic energy output operating without electrolyte replenishing. **e** Long-term power output stability and corresponding structure stability of the hydrated 2D-NNF reflecting from the lamellar nanochannel space. **f** Comparisons of the key performance parameters between the 2D-NNF and reported state-of-the-art analogs (GO and MXene-based ones) for evaluating large-scale osmotic energy harvesting. The error bars in (**c**) and (**e**) represent the standard deviations for three measurements. Source data are provided as a Source Data file.

summarized to evaluate the resource, environment, and economic impacts of 2D-NNF developed in this work, which included the steps from raw material processing to laboratory exfoliation, and analyzed the material flow in detail[68] (Fig. 6a,b and Supplementary Table 7). The specific calculation details of the technoeconomic analysis are provided in the Supplementary Note 8. Taking the final production of 5 kg of material as an example, the feed amount of each material was estimated according to the yield ratio of each stage. In contrast to the chemical synthesis of GO and MXene, the production process of MMT primarily involves physical methods, thus high-purity MMT materials were obtained by tertiary hydraulic grading and centrifugal purification, and then few-layer nanosheets were obtained by mechanical stirring (Fig. 6a). Regarding the nano cellulose, the environmental impact of CNF was 1/10 of that of ANF at an equivalent mass (Supplementary Table 8), despite the involvement of a small amount of chemical reagents in the preparation process.

From a resource and environment impact perspective, the production of GO nanofiber-based nanofluidics necessitated the use of strong corrosive and oxidizing reagents, leading to an extremely high global warming potential (GWP) gases, such as $CO_2$ and $CH_4$ (Fig. 6c). Furthermore, the extensive use of chemical reagents also increased the emissions of acidic gases such as $SO_2$ (acidification potential, AP) in the whole life cycle and resulted in a high equivalent freshwater eco-toxicity potential (FAETP). In the case of MXene-based nanofluidics, the production process involved ball milling and high-temperature calcination, resulting in significant energy consumption. Moreover, the use of LiF and HCl during the exfoliation process will lead to a high human toxicity potential (HTP) and marine aquatic eco-toxicity potential (MAETP) equivalents, which posed significant threats to human and marine ecology. From an economic standpoint, the complex preparation process of GO and the high cost of MXene raw

materials, specifically metal powder, further hindered their practical application.

In contrast, the production process of the MMT nanofiber-based membrane (2D-NNF) primarily utilizes physical methods for substance purification without the use of strong corrosive and oxidizing reagents. Consequently, the energy consumption and abiotic depletion (ADP fossil) in the production process of 2D-NNF are only 2/15 and 1/14 of GO and MXene nanofiber-based membrane, respectively. Additionally, the corresponding impact in GWP, AP, HTP, MATEP, and FAETP are 1/9, 1/13, 1/61, 1/45, and 1/14, respectively (Supplementary Table 10). Significantly, the 2D-NNF exhibited an extremely low cost (1.0 USD m$^{-2}$) advantage by comparison with the mainstream 2D material (i.e., GO/ANF: 3.8 USD m$^{-2}$, MXene/CNF: 12.6 USD m$^{-2}$) nanofiber-based nanofluidics (Fig. 6c, Supplementary Tables 9 and 10). It is well known that the economic cost is a crucially important factor for the industrial prospect. The cost for 1 m$^2$ 2D-NNF was quite lower than the commercially accepted target (4.8 USD m$^{-2}$)[62,69].

However, it should be noted that the water consumption of MMT in the material flow analysis is larger than that of MXene (Fig. 6a). In order to describe the influence of different nanofluidics more objectively, water consumption was taken as the representative of resource indicators, GWP and cost as the representative of environmental and economic indicators, and the sensitivity analysis of the evaluation model was carried out by TOPSIS (Technique for Order Preference by Similarity to an Ideal Solution) method (Supplementary Note 9)[70,71]. As shown in Fig. 6d, when the weights were changed to allocate among the indicators, the sorting result remained unchanged in more than 88% of cases, that is, 2D-NNF was still the optimal choice. Therefore, the fabrication of 2D-NNF is simple, scalable, environmentally friendly, and pollution-free, and can be applied to different functional types of ion regulation. Thus, it promised a substantial payback by applying the designed nanofluidics for osmotic energy harvesting.

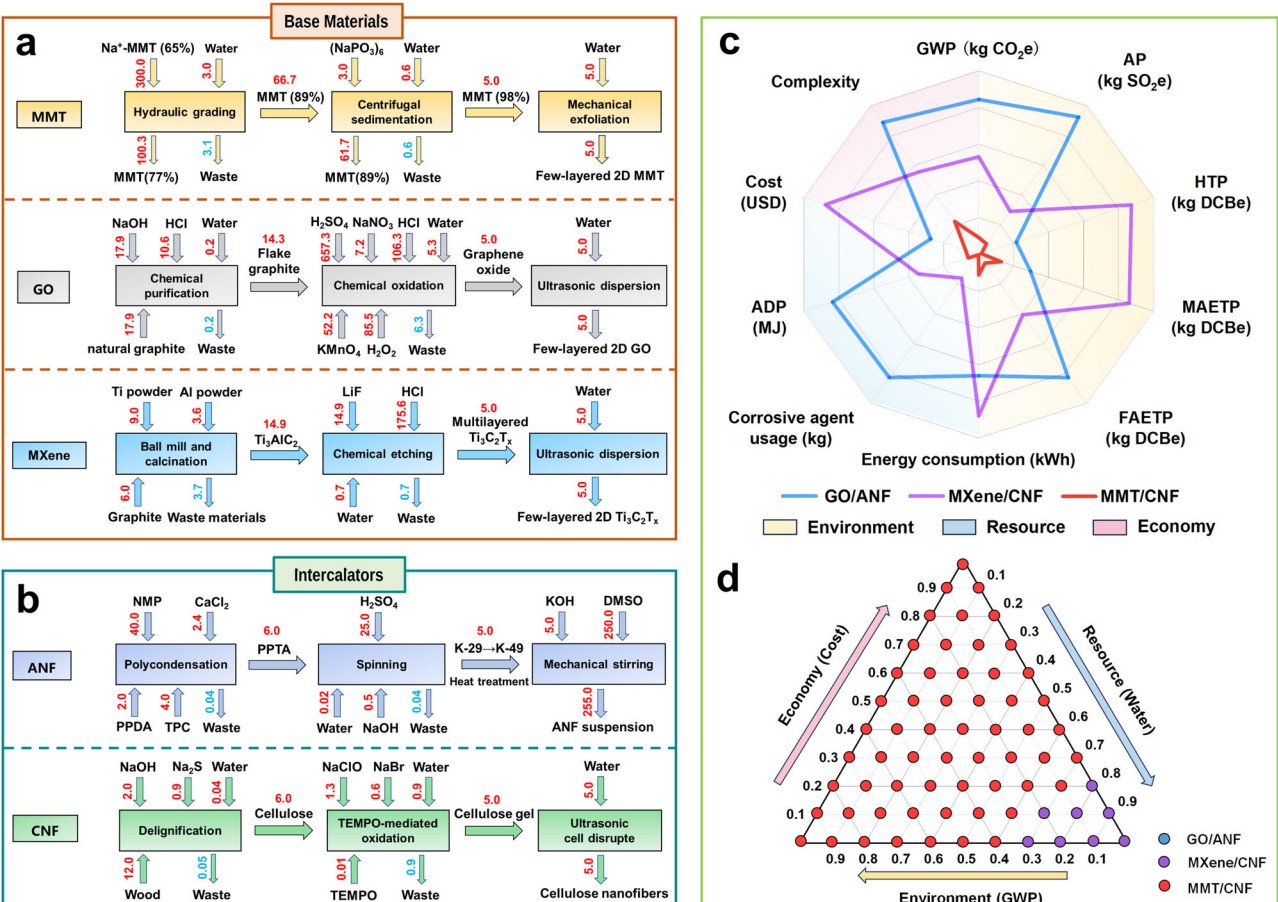

**Fig. 6 | Resource, environmental, and technoeconomic analysis.** Material flow analysis of the base material of MMT, GO, and MXene (**a**), and the intercalator of ANF and CNF (**b**). The red numbers in the arrows represent the mass of each material. NMP 1-methyl-2-pyrrolidinone, PPDA p-Phenylenediamine, TPC terephthaloyl chloride, PPTA poly-p-phenylene terephthamide, K-29 kevlar-29, DMSO dimethyl sulfoxide, TEMPO 2,2,6,6-tetramethyl-1-piperinedinyloxy. Except water (*t*), the unit of other auxiliary materials is kg. **c** Comparisons of impact indexes of the all-natural 2D nanofluidics developed in this work with reported state-of-the-art analogs from the economy (pink area), environment (yellow area), and resource (blue area) perspectives. GWP represents global warming potential, AP represents acidification potential, HTP represents human toxicity potential, MAETP represents marine aquatic eco-toxicity potential, FAETP represents freshwater eco-toxicity potential, and ADP represents abiotic depletion. **d** Sensitivity analysis of the influence of index weight change on the stability of evaluation model.

## Discussion

In summary, we developed robust 2D-NNF by utilizing natural clay minerals (MMT) and natural celluloses (CNFs) for achieving a high-efficiency blue osmotic energy harvesting above the industrial level. The interlocking between MMT base and CNF intercalator contributed to a high mechanical strength of the obtained membrane and thus benefited for long-term stability in the application of osmotic energy generation. The abundant negative charges on the MMT and CNFs also generated a strong electrostatic field in the interlamellar nanochannels, which promoted the selective and rapid cation transport for highly efficient osmotic energy harvesting. Serving as an osmotic energy generator, the 2D-NNF could output a maximum power up to 8.61 W m$^{-2}$ between artificial seawater and river water, much higher than reported 2D nanofluidics. When the area of the 2D-NNF is enlarged to 700 cm$^2$, the different sites selected from various regions of membrane could still exhibit high power density generation (~8.36 W m$^{-2}$) and long-term stability (>30 days). More importantly, the resource, environmental and technoeconomic analysis further showed that the production process of 2D-NNF significantly reduced resource consumption (1/14), greenhouse gas emissions (1/9), and production costs (1/13) in comparison to the GO and MXene-based membrane. This work advances the large-scale applications of all-natural 2D nanofluidics in blue osmotic energy harvesting by utilizing sustainable raw materials, which also provide great potential in ion sieving and desalination.

## Methods

### Materials

Na-Montmorillonite (MMT) was purchased from Nanocor in the United States. TEMPO-oxidized CNFs gel (1 wt%) was purchased from Tianjin Woodelfbio Cellulose Co., Ltd. Mixed cellulose (MCE) filter membrane (pore size -0.22 μm) was provided by Tianjin Jinteng Co., Ltd. All of the chemicals including potassium chloride (KCl), sodium chloride (NaCl), lithium chloride (LiCl), calcium chloride (CaCl$_2$) and magnesium chloride hexahydrate (MgCl$_2$.6H$_2$O) were analytically pure.

### Fabrication of the 2D natural nanofluidics (2D-NNF)

Firstly, CNFs gel was dispersed in deionized water and then ultrasonication for 20 min to produce a CNFs suspension at 1 mg/mL. MMT was stirred and ultrasonic dispersed in deionized water to get a uniform suspension at 4 mg/mL. Subsequently, the MMT dispersion was mixed with a certain amount of CNF suspension, and the mixture was further stirred for 2 h and sonicated for 20 min. In order to form a more uniform dispersion, the synthesized mixture needs to stand at room temperature for more than 24 h. Then, the 2D-NNF mixture was vacuum filtration on the MCE filter film and followed by drying in the air for 12 h. Finally, the 2D-NNF can be easily peeled from the MCE

substrate. For large-area nanofluidics preparation, MMT and CNF dispersion should be maintained at 0.5 mg/mL.

## Electrical measurements

The as-prepared 2D-NNF was mounted between a two-chamber electrochemical cell to test the ion transport properties and osmotic energy conversion performance. A pair of homemade Ag/AgCl electrodes were used to apply a transmembrane potential. It should be noted that the effective area of Ag/AgCl electrode was about 1.36 cm². And the effective membrane testing area was set about 0.03 mm². The current-voltage (*I-V*) measurements and energy collecting tests were performed by a Keithley 6487 picoammeter (Keithley Instruments), and the specific test method was shown in Supplementary Fig. 20. The diluted HCl and KOH solutions were used to adjust the pH value of the electrolyte. The testing solutions were used to repeatedly wash the cell before each measurement. And the testing solutions were all prepared using ultrapure water (18.2 MΩ cm).

## Characterizations

The morphology of the samples was obtained by scanning electron microscopy (SEM, Gemini SEM 300) and transmission electron microscopy (TEM, JEM-2100F). The interlayer spacing XRD analysis was carried out using a Bruker D8 Advance with filtered Cu-Kα radiation ($\lambda = 0.154$ nm) and was recorded at the range of 2θ = 2°–10°. The AFM images were obtained by Bruker Dimension Icon with tapping mode. FTIR spectroscopy measurements were conducted by a FTIR spectrometer (Perkin Elmer Frontier) in the wavenumber range of 400–4000 cm$^{-1}$. The XPS analysis was performed using an ESCALAB 250Xi spectrometer (Thermo Fisher Scientific) with monochromated Al-Kα radiation. The zeta potential of MMT and 2D-NNF solutions was measured by Zeta potential analyzer (Nano ZS90, Malvern, UK). For the mechanical testing, the membranes were cut into strips (30 mm × 5 mm). The tensile tests were performed at a loading rate of 1 mm/min at room temperature by using a Shimadzu AGS-X tester. The wettability test was carried out on a contact angle measuring equipment (JC2000D1) at room temperature.

## Statistical analysis

The linear correlation between two variables was assessed using Pearson correlation analysis. One-way student *t*-test was used to analyze data that were normally distributed, and the values were presented as the mean ± standard deviation. The data were analyzed using SPSS version 19, and a *P* value less than 0.05 was considered significant.

## Technique for order preference by similarity to an ideal solution, TOPSIS

The initialization decision matrix was established and normalized. By calculating the distance between each scheme and the optimal scheme and the worst scheme, the closeness between each scheme and the optimal scheme is obtained, and the scheme was sorted according to this. The detailed calculations were provided in Supplementary Note 9.

## Reporting summary

Further information on research design is available in the Nature Portfolio Reporting Summary linked to this article.

## Data availability

All data supporting the research in this study are available within the article and supplementary information file. Source data are provided with this paper (ref. 72). Source data are provided with this paper.

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

## Acknowledgements

This work was supported by the National Natural Science Foundation of China (No. 62075002, by Q.Z., No. 22033006, by Z.Z.), Beijing Nova Program (No. 20220484234, by Q.Z., No. 20230484265, by Y.G.) and Natural Science Foundation of Beijing Municipality (No. 2212001, by Q.Z.). The authors thank Prof. Longcheng Gao from Beihang University for the help in this work.

## Author contributions

J.T. completed a major part of the experiments and drafted the original version of the manuscript. Y.W. completed the theoretical calculation part using MD simulations and drafted part of the manuscript. Y.G. and H.Y. developed the resource, environmental, and technoeconomic analysis model. Q.Z., Z.Z., Y.G., and T.Z. discussed together and proposed the concepts of this paper and made extensive revisions to the original manuscript. C.W., Y.J., and H.W. assisted in the design of some test methods and materials characterization. L.L. drew some of the pictures for the paper. All authors contributed to this work.

## Competing interests

The authors declare that they have no competing interests.
