## [Peer Review File · Nature Communications]

All-natural 2D nanofluidics as highly-efficient osmotic energy generatorsREVIEWER COMMENTS

Reviewer #1 (Remarks to the Author):

I have carefully read the manuscript entitled 'All-natural 2D nanofluidics as high-efficient osmotic energy generator'. The manuscript presents the development of a two-dimension all-natural nanofluidic (2D-NNF) is developed as robust and high-efficient osmotic energy generator based on an interlocking configuration of the stacked montmorillonite nanosheets (from natural clay) and their intercalated cellulose nanofibers (from natural wood). The manuscript is very well written and organised with a high amount of analysis methods that support the conclusions and experimental hypothesis. Before acceptance and publication, I strongly suggest to add a centralised table in order to compare previous reported results in literature.

Reviewer #2 (Remarks to the Author):

In this work, Tang et al. developed a kind of porous membrane with natural materials for osmotic energy conversion. With an interlocking configuration between the montmorillonite matrix and the intercalated cellulose nanofibers, the fabricated porous membrane presents strong mechanical strength and tunable widths of interlayer nanochannels. With the confined nanochannels and negative surface charges, the membrane has strong cation selectivity which is a good candidate for the high-performance osmotic energy conversion. The authors conducted MD simulations to provide the physical details of ionic diffusion across the porous membranes. Detailed life cycle assessment was also done to show the economic, environmental, and resource impacts of the developed membrane.

The manuscript can be improved by solving the following questions:

1. The manuscript was not prepared carefully. Many grammatical errors in the text should be revised carefully, such as:

Line 39

a salinity gradient

Line 140

'were' should be 'was'

Line 269

gradients

Line 337

an important factor

In the whole manuscript, 'pH' should be used correctly.

There are also some poor statements, like:

Line 293-295

The experimental conditions should be provided. Or it's hard to follow the description.

Line 421 'According to the literature reports'

While, there are no references provided in the sentence.

2. Line 28, Line 99, and Line 488

In this work, the authors prepared membranes with a diameter of 30 cm. However, the original membrane was not used directly for the osmotic energy conversion. The description in the three places should be changed to eliminate misunderstandings about osmotic energy conversion with the 30-cm-diameter membrane.

3. Line 129

During the fabrication of the porous membrane, are there any experimental procedures that ensure the roughly uniform film thickness/ porosity over a larger area? How many membranes had been prepared in this work? Does the thickness (or other physical parameters) demonstrate good repeatability?

4. Line 136

The nanochannel shows expansion in the fully hydrated state. Does the expansion depend on the applied solutions which have different Debye lengths?

5. Line 141

The authors claimed that the surface charges originate from the deprotonation of carboxyl and hydroxyl groups. (Line 117) During the preparation of the membrane, is the aqueous solution the same in concentration and pH?

As shown in Figure S7, why does the surface charge density increase with the salt concentration? Under 1 M KCl, the surface charge density can reach ~ -65.6 mC/m².

Also, why does the surface charge density have no dependence on the solution pH? Is there any consideration of the concentration of H⁺ and OH⁻ when using equation S2, especially at a very large or small pH?

6. Line 184-186

The hopping mechanism is usually used to describe the transport of protons. The authors need to provide more statements about the applicability of the hopping mechanism in the diffusion process of K ions. In aqueous solutions, K ions are hydrated and much larger than protons.

As shown by the inset of Fig.2a, is the ionic hopping along the horizontal direction in Fig.2f?

However, the MD simulations show similar velocities of K and Cl ions. More discussions can be provided.

7. Line 212

The nanochannels in the membranes have sub-2 nm widths (Fig.1e). As shown in Fig S1b, the nanofibers have an averaged diameter of ~ 6 nm, much larger than 2 nm. Then, how can the nanofibers tune the surface charge density of MMT channels? A scheme may be needed to show the structure of the membrane. The microscale structure of the membrane is important which determines the conduction of MD simulations and the ionic behaviors.

8. Line 214

More details about the MD simulations should be provided, like how many water molecules, cations, anions, and charges in the simulation systems.

9. Line 239

12 ns may be not long enough to obtain analyzable data. From Fig. 2e, the number of cations diffusing through the membrane is only ~ 10 .

The authors can explain why the diffused ion number shows a large decrease at ~ 6 ns.

10. For equation S5, the ion activity coefficients should be provided for the solutions used in the work.

11. Line 291

For the natural situation at the estuaries, the concentration of the seawater is ~ 0.5 M. The sentence '2D-NNF could still work in high salt solution' makes me puzzled.

12. Line 352-353

At pH 11, the surface potential seems to remain the same from Fig. S7. The pK_a value for carboxyl groups is ~ 3.8 , in solutions with pH higher than 8 or 9, all groups should be deprotonated. The surface charge density can reach its largest value. Maybe more discussion can be added here.

13. Line 366

From the nanofluidic experiments, the fabricated membrane exhibits high-performance osmotic energy conversion. Is it possible to conduct the osmotic energy generation experiment with the 30-cm-diameter membrane directly? If so, this is much closer to the real application.

Reviewer #3 (Remarks to the Author):

The authors developed a highly efficient osmotic energy generator using natural clay-based nanofluidics. They created a two-dimensional nanofluidic system by combining montmorillonite nanosheets from clay and cellulose nanofibers from wood, enabling rapid cation transport and achieving an impressive osmotic power output of 8.61 W.m⁻², surpassing previous 2D nanofluidic systems. The membrane's scalability and stability were also demonstrated, making it suitable for practical osmotic energy harvesting. I found the study intriguing and believe it enhances our understanding of osmotic energy production. However, there are concerns that the authors need to comprehensively address. While their paper shows promise, a thorough revision is necessary before it can be accepted for publication.

1. The graphical abstract is one of the main pillars of the article, and at the same time, it should include the main topic. Authors need to include more details about nanochannels in the graphical abstract to make it more informative.

2. It is suggested to give further details of the findings in the abstract.

3. The text does not mention specific electrokinetic phenomena in nanofluidic systems such as ionic current rectification, ionic concentration polarization, energy harvesting, and reduced fouling. To this end, the following relevant references must be cited properly in the introduction and throughout the manuscript:

- <https://doi.org/10.1039/D0CP05974A>
- <https://doi.org/10.1016/j.electacta.2021.139221>
- <https://doi.org/10.1039/D2CP01015A>
- <https://doi.org/10.1016/j.electacta.2022.141175>
- <https://doi.org/10.1021/acs.langmuir.2c01790>

4. The leading cause of mass transfer in nanochannels is the formation and distortion of electrical double layers (EDLs). The text of the manuscript has almost nothing on this issue. Authors must adequately clarify EDL theories and the factors that influence them. To this end, the authors can cite:

- <https://doi.org/10.1016/j.electacta.2021.139376> ,
- <https://doi.org/10.1021/acs.analchem.2c04559> ,
- <https://doi.org/10.1063/5.0160678>

5. The authors should elaborate on the interactions between montmorillonite nanosheets and cellulose nanofibers at the molecular level. How do these interactions contribute to the structural stability and functionality of the membrane?

6. What specific mechanisms facilitate the rapid transport of cations within the developed two-dimensional nanofluidic system, particularly considering the interplay between montmorillonite nanosheets and cellulose nanofibers?

7. How were the surface and space negative charges effectively incorporated into the interlamellar channels, and what role did these charges play in enhancing the selective transport of cations, leading to the significant osmotic power output? It is supposed that the authors comment on these issues.

8. The authors should provide more details about the parameters and methodologies used in the life cycle assessment. What were the key findings regarding the economic, environmental, and energy benefits in comparison with other existing osmotic energy generation methods?

9. What strategies were employed to ensure the long-term stability of the large-area membrane over the course of 30 days? What potential challenges and solutions were encountered during this period?

10. In practical terms, how versatile is this technology across different environmental conditions? Are there specific environmental factors (temperature, salinity, etc.) that might influence its efficiency and stability in real-world osmotic energy harvesting applications?

11. Please edit the language carefully, fix typos, and correct grammatical errors.

RESPONSE TO REVIEWERS' COMMENTS

The authors appreciate the reviewers for these very helpful comments. Please find our point-by-point responses as follows:

Responses to Reviewer #1:

I have carefully read the manuscript entitled 'All-natural 2D nanofluidics as high-efficient osmotic energy generator'. The manuscript presents the development of a two-dimension all-natural nanofluidic (2D-NNF) is developed as robust and high-efficient osmotic energy generator based on an interlocking configuration of the stacked montmorillonite nanosheets (from natural clay) and their intercalated cellulose nanofibers (from natural wood). The manuscript is very well written and organised with a high amount of analysis methods that support the conclusions and experimental hypothesis.

Before acceptance and publication, I strongly suggest to add a centralised table in order to compare previous reported results in literature.

Response: Thank you for the encouraging and helpful comments concerning our manuscript. To demonstrate the performance level of 2D-NNF developed in this work, we have conducted a careful literature research on the previously reported nanofluidics (including 1D, 2D and 3D) and summarized key performance parameters into a table according to your helpful suggestions. The comparison results show that our 2D-NNF has a superior osmotic energy harvesting behavior (power density of 8.61 W m^{-2}) compared with all previously reported 2D nanofluidics (**Table R1**). However, the power density of 2D-NNF is lower than some state-of-the-art 1D and 3D nanofluidics due to the generally accepted ion transport paths differences caused by distinct nanochannel configurations, which provides the further research directions for optimum structural design of 2D-NNF. For an adequate comparison, we have added this table and above discussions in our revised manuscript (**Page 12, Table S5**). The performance comparisons (**Fig. 3f, Page 11**) have also been updated based on the centralised table. We hope our revised version could meet the standards of this journal.

Table R1 Comparisons for key performance parameters of the 2D-NNF developed in this work with previously reported single 1D, 2D and 3D nanofluidics. All the measurements were carried out by mixing artificial seawater and river water (0.5 M / 0.01 M NaCl).

Material system	Con-figuration	Power density (W/m²)	Resistance (kΩ)	Thickness (μm)	Test area (mm²)	Ref.
2D-NNF	2D	8.61	6	5.3	0.03	This work
Heterogeneous MXene	2D	8.6	10	4	0.03	1
iGOM	2D	6.78	30	10	0.03	2
GO/IL	2D	6.7	13	11	0.03	3
WS ₂ /ANF	2D	6.01	23	4	0.03	4
NBCM	2D	5.58	30	12	0.03	5
BN/ANF	2D	5.5	10	1	0.03	6
bsGOM	2D	5.5	36	4	0.03	7
Asymmetric GO	2D	5.32	30	5.87	0.04	8
MoS ₂ /CNF	2D	5.2	24	4	0.03	9
GO/SNF/GO	2D	5.07	40	5	0.03	10
GO/ANF	2D	5.06	20	2.2	0.03	11
Fusiform-GOMs	2D	4.94	26	2	0.03	12
MXene/CNF	2D	4.84	13	5	0.03	13
GO/CNF	2D	4.19	27	9	0.03	14
STFA-Na-MTM	2D	4.10	17	9.3	0.03	15
VMT	2D	4.1	15	2.1	0.03	16
GO/BP	2D	3.4	200	8	0.03	17
MXene/ANF	2D	3.7	27	4.5	0.03	18
MMT/ANF	2D	3.38	8	3	0.03	19
LDH/AAO	2D	2.85	23	45.8	0.03	20
MXene/BN	2D	2.3	60	10	0.03	21
Heterogeneous MXene	2D	2.3	15	3.5	0.03	22

Material system	Con-figuration	Power density (W/m ²)	Resistance (kΩ)	Thickness (μm)	Test area (mm ²)	Ref.
WS ₂ /CNF	2D	1.99	13	3	0.03	23
BP	2D	1.6	250	8	0.03	17
C-MXene/C-HNF	2D	1.09	17	—	0.03	24
Heterogeneous MUM	2D	0.7	30	16.5	0.2	25
GO/PPSU-Py	2D	0.76	300	31	0.03	26
MXene	2D	0.53	11	15	0.03	27
COF-(SO ₃ Na) ₁ /PAN	1D	97	8.2	0.053	0.00785	28
p-BCP-1	1D	19.3	10	—	0.008	29
TpEB@TpPa-SO ₃ Na	1D	19.2	5.5	0.5	0.03	30
ZnTPP-COF	1D	14.63	7	0.0011	0.006	31
TFPT-TMT COF	1D	13.3	17	0.3	0.00785	32
h-PEI	1D	13.2	15	0.012	0.008	33
PyPa-SO ₃ H COF/SANF	1D	9.6	5	7.2	0.03	34
PS-b-P2VP/MXene	1D	6.74	5.5	1.1	0.03	35
TpPa-SO ₃ H COF	1D	5.9	23	10.7	0.03	36
Cation-selective COF/AAO	1D	5.41	13	25	0.03	37
PAA-cPEI	1D	3.7	40	0.214	0.008	38
BDA-TAM	1D	2.96	17	1.5	0.02	39
Anion-selective COF/AAO	1D	2.5	5	38	0.03	40
BCP	1D	2.1	17	0.5	0.03	41
NMIM	3D	23	5	200	0.03	42
SPEEK/AAO/PPy	3D	9.65	2	2	0.03	43
MOF-on-MOF	3D	8.72	10	75.7	0.03	44
SPEEK-SPSF	3D	7	17	4.3	0.03	45
ZW-M7N1	3D	6.2	17	65	0.03	46
ICM-4	3D	6.18	20	—	0.03	47

Material system	Con-figuration	Power density (W/m²)	Resistance (kΩ)	Thickness (μm)	Test area (mm²)	Ref.
Anti-Swelling hydrogel	3D	6	10	50	0.03	48
HEMAP hydrogel	3D	5.38	15	25	0.03	49
PSS/HKUST-1/AAO	3D	5.2	—	0.2	0.03	50
MCS/AAO	3D	5.04	10	—	0.03	51
KANF	3D	4.8	10	4	0.03	52
ANF/Gel	3D	3.9	23	210	0.03	53
Carbon/AAO	3D	3.46	10	64	0.03	54
Hydrogel hybrid	3D	3.18	36	25	0.03	55
SNF/AAO	3D	2.86	23	65	0.03	56
CMWs	3D	2.78	13	135	0.03	57
PAEK-HS/PES-Py	3D	2.66	10	11	0.03	58
HENM	3D	2.22	10	—	0.03	59
TPPS/Al ₂ O ₃	3D	2.16	10	25.2	0.03	60

Reference

- 1 Ding, L. *et al.* Bioinspired $Ti_3C_2T_x$ MXene-based ionic diode membrane for high-efficient osmotic energy conversion. *Angew. Chem. Int. Ed.* **61**, e202206152 (2022).
- 2 Yan, P. P. *et al.* Two-dimensional nanofluidic membranes with intercalated in-plane shortcuts for high-performance blue energy harvesting. *Small* **19**, 2205003 (2023).
- 3 Hu, Y. *et al.* Confined ionic-liquid-mediated cation diffusion through layered membranes for high-performance osmotic energy conversion. *Adv. Mater.*, 2301285 (2023).
- 4 Wang, Q. *et al.* Efficient solar-osmotic power generation from bioinspired anti-fouling 2D WS_2 composite membranes. *Angew. Chem. Int. Ed.* **62**, e202302938 (2023).
- 5 Zhang, M. *et al.* Enhanced selective ion transport by assembling nanofibers to membrane pairs with channel-like nanopores for osmotic energy harvesting. *Nano Energy* **103**, 107786 (2022).
- 6 Chen, C. *et al.* Bio-inspired nanocomposite membranes for osmotic energy harvesting. *Joule* **4**, 247-261 (2020).
- 7 Qian, Y. *et al.* Boosting osmotic energy conversion of graphene oxide membranes via self-exfoliation behavior in nano-confinement spaces. *J. Am. Chem. Soc.* **144**, 13764-13772 (2022).
- 8 Bang, K. R., Kwon, C., Lee, H., Kim, S. & Cho, E. S. Horizontally asymmetric nanochannels of graphene oxide membranes for efficient osmotic energy harvesting. *ACS Nano* **17**, 10000-10009 (2023).
- 9 Zhu, C. *et al.* Metallic two-dimensional MoS_2 composites as high-performance osmotic energy conversion membranes. *J. Am. Chem. Soc.* **143**, 1932-1940 (2021).
- 10 Xin, W. *et al.* Biomimetic nacre-like silk-crosslinked membranes for osmotic energy harvesting. *ACS Nano* **14**, 9701-9710 (2020).
- 11 Chen, J. *et al.* Biomimetic nanocomposite membranes with ultrahigh ion selectivity for osmotic power conversion. *ACS Cent. Sci.* **7**, 1486-1492 (2021).
- 12 Qian, Y. *et al.* Two-dimensional membranes with highly charged nanochannels for osmotic energy conversion. *ChemSusChem* **15**, e202200933 (2022).
- 13 Liu, P. *et al.* Synergy of light and acid-base reaction in energy conversion based on cellulose nanofiber intercalated titanium carbide composite nanofluidics. *Energy Environ. Sci.* **14**,

- 4400-4409 (2021).
- 14 Wu, Y. *et al.* Enhanced ion transport by graphene oxide/cellulose nanofibers assembled membranes for high-performance osmotic energy harvesting. *Mater. Horiz.* **7**, 2702-2709 (2020).
- 15 Ding, Z. *et al.* Promoting osmotic energy conversion through fluorinated nanochannel membranes with large-scale exfoliation and low transmission resistance. *J. Mater. Chem. A.* **11**, 8798-8808 (2023).
- 16 Cao, L. *et al.* Lamellar porous vermiculite membranes for boosting nanofluidic osmotic energy conversion. *J. Mater. Chem. A.* **9**, 14576-14581 (2021).
- 17 Zhang, Z. *et al.* Oxidation promoted osmotic energy conversion in black phosphorus membranes. *Proc. Natl. Acad. Sci. U.S.A.* **117**, 13959-13966 (2020).
- 18 Zhang, Z. *et al.* Mechanically strong MXene/Kevlar nanofiber composite membranes as high-performance nanofluidic osmotic power generators. *Nat. Commun.* **10**, 1-9 (2019).
- 19 Qin, R. *et al.* Nanofiber-reinforced clay-based 2D nanofluidics for highly efficient osmotic energy harvesting. *Nano Energy* **100**, 107526 (2022).
- 20 Liu, Y., Ping, J. & Ying, Y. Anion-selective layered double hydroxide composites-based osmotic energy conversion for real-time nutrient solution detection. *Adv. Sci.* **9**, 2103696 (2022).
- 21 Yang, G. *et al.* Stable $Ti_3C_2T_x$ MXene–boron nitride membranes with low internal resistance for enhanced salinity gradient energy harvesting. *ACS Nano* **15**, 6594-6603 (2021).
- 22 Wang, J. *et al.* Heterogeneous two-dimensional lamellar $Ti_3C_2T_x$ membrane for osmotic power harvesting. *Chem. Eng. J.* **452**, 139531 (2023).
- 23 Gao, Z. *et al.* Design of metallic phase WS_2 /cellulose nanofibers composite membranes for light-boosted osmotic energy conversion. *Carbohydr. Polym.* **296**, 119847 (2022).
- 24 Rao, J. *et al.* Nacre-inspired mechanically robust films for osmotic energy conversion. *Adv. Funct. Mater.*, 2309869 (2023).
- 25 Wei, C. *et al.* Parallel arrays of clay nanosheets sandwiched in two-dimensional nanofluidic membrane for enhanced ion transport properties. *J. Membr. Sci.* **680**, 121744 (2023).
- 26 Zhu, X. *et al.* A charge-density-tunable three/two-dimensional polymer/graphene oxide

- heterogeneous nanoporous membrane for ion transport. *ACS Nano* **11**, 10816-10824 (2017).
- 27 Liu, P. *et al.* Neutralization reaction assisted chemical-potential-driven ion transport through layered titanium carbides membrane for energy harvesting. *Nano Lett.* **20**, 3593-3601 (2020).
- 28 Zuo, X. *et al.* Thermo-osmotic energy conversion enabled by covalent-organic-framework membranes with record output power density. *Angew. Chem. Int. Ed.* **61**, e202116910 (2022).
- 29 Li, C. *et al.* One porphyrin per chain self-assembled helical ion-exchange channels for ultrahigh osmotic energy conversion. *J. Am. Chem. Soc.* **144**, 9472-9478 (2022).
- 30 Cao, L. *et al.* An ionic diode covalent organic framework membrane for efficient osmotic energy conversion. *ACS Nano* **16**, 18910-18920 (2022).
- 31 Yang, J. *et al.* Advancing osmotic power generation by covalent organic framework monolayer. *Nat. Nanotechnol.* **17**, 622-628 (2022).
- 32 Wang, K. *et al.* Monolayer-assisted surface-initiated schiff-base-mediated aldol polycondensation for the synthesis of crystalline sp² carbon-conjugated covalent organic framework thin films. *J. Am. Chem. Soc.* **145**, 5203-5210 (2023).
- 33 Li, C. *et al.* Large-scale, robust mushroom-shaped nanochannel array membrane for ultrahigh osmotic energy conversion. *Sci. Adv.* **7**, eabg2183 (2021).
- 34 Man, Z. *et al.* Serosa-mimetic nanoarchitecture membranes for highly efficient osmotic energy generation. *J. Am. Chem. Soc.* **143**, 16206-16216 (2021).
- 35 Lin, X. *et al.* Heterogeneous MXene/PS-b-P2VP nanofluidic membranes with controllable ion transport for osmotic energy conversion. *Adv. Funct. Mater.* **31**, 2105013 (2021).
- 36 Hou, S. *et al.* Free-standing covalent organic framework membrane for high-efficiency salinity gradient energy conversion. *Angew. Chem. Int. Ed.* **133**, 10013-10018 (2021).
- 37 Gao, M. *et al.* A bioinspired ionic diode membrane based on sub-2 nm covalent organic framework channels for ultrahigh osmotic energy generation. *Nano Energy* **105**, 108007 (2023).
- 38 Yang, X. *et al.* Enhanced osmotic energy conversion through an asymmetric nanochannel array membrane with an ultrathin selective layer. *Chem. Mater.* **35**, 7266-7272 (2023).
- 39 Wang, C. *et al.* Ultrathin self-standing covalent organic frameworks toward highly-efficient

- nanofluidic osmotic energy generator. *Adv. Funct. Mater.* **32**, 2204068 (2022).
- 40 Chen, M. *et al.* In situ growth of imine-bridged anion-selective COF/AAO membrane for ion current rectification and nanofluidic osmotic energy conversion. *Adv. Funct. Mater.*, 2302427 (2023).
- 41 Zhang, Z. *et al.* Ultrathin and ion-selective janus membranes for high-performance osmotic energy conversion. *J. Am. Chem. Soc.* **139**, 8905-8914 (2017).
- 42 Zhang, F., Yu, J., Si, Y. & Ding, B. Meta-aerogel ion motor for nanofluid osmotic energy harvesting. *Adv. Mater.*, 2302511 (2023).
- 43 Hao, J. *et al.* A euryhaline-fish-inspired salinity self-adaptive nanofluidic diode leads to high-performance blue energy harvesters. *Adv. Mater.* **34**, 2203109 (2022).
- 44 Tonnah, R. K. *et al.* Bioinspired angstrom-scale heterogeneous MOF-on-MOF membrane for osmotic energy harvesting. *ACS Nano* **17**, 12445-12457 (2023).
- 45 Zhao, X. *et al.* Metal organic framework enhanced SPEEK/SPSF heterogeneous membrane for ion transport and energy conversion. *Nano Energy* **81**, 105657 (2021).
- 46 Sun, Y. *et al.* Tailoring a poly (ether Sulfone) bipolar membrane: osmotic-energy generator with high power density. *Angew. Chem. Int. Ed.* **132**, 17576-17581 (2020).
- 47 Chen, W. *et al.* Ionic crosslinking-induced nanochannels: nanophase separation for ion transport promotion. *Adv. Mater.* **34**, 2108410 (2022).
- 48 Bian, G. *et al.* Anti-swelling gradient polyelectrolyte hydrogel membranes as high-performance osmotic energy generators. *Angew. Chem. Int. Ed.* **133**, 20456-20462 (2021).
- 49 Chen, W. *et al.* Improved ion transport and high energy conversion through hydrogel membrane with 3D interconnected nanopores. *Nano Lett.* **20**, 5705-5713 (2020).
- 50 Pan, S. *et al.* Toward scalable nanofluidic osmotic power generation from hypersaline water sources with a metal-organic framework membrane. *Angew. Chem. Int. Ed.* **62**, e202218129 (2023).
- 51 Zhou, S. *et al.* Interfacial super-assembly of ordered mesoporous carbon-silica/AAO hybrid membrane with enhanced permselectivity for temperature-and pH-Sensitive smart ion transport. *Angew. Chem. Int. Ed.* **133**, 26371-26380 (2021).
- 52 Ding, L. *et al.* Ultrathin and ultrastrong kevlar aramid nanofiber membranes for highly stable osmotic energy conversion. *Adv. Sci.* **9**, 2202869 (2022).

- 53 Zhang, Z. *et al.* Improved osmotic energy conversion in heterogeneous membrane boosted by three-dimensional hydrogel interface. *Nat. Commun.* **11**, 1-8 (2020).
- 54 Gao, J. *et al.* High-performance ionic diode membrane for salinity gradient power generation. *J. Am. Chem. Soc.* **136**, 12265-12272 (2014).
- 55 Chen, W. *et al.* Improved ion transport in hydrogel-based nanofluidics for osmotic energy conversion. *ACS Cent. Sci.* **6**, 2097-2104 (2020).
- 56 Xin, W. *et al.* High-performance silk-based hybrid membranes employed for osmotic energy conversion. *Nat. Commun.* **10**, 1-10 (2019).
- 57 Xie, L. *et al.* Sequential superassembly of nanofiber arrays to carbonaceous ordered mesoporous nanowires and their heterostructure membranes for osmotic energy conversion. *J. Am. Chem. Soc.* **143**, 6922-6932 (2021).
- 58 Zhu, X. *et al.* Unique ion rectification in hypersaline environment: A high-performance and sustainable power generator system. *Sci. Adv.* **4**, eaau1665 (2018).
- 59 Ling, H. *et al.* Heterogeneous electrospinning nanofiber membranes with pH-regulated ion gating for tunable osmotic power harvesting. *Angew. Chem. Int. Ed.* **135**, e202212120 (2023).
- 60 Zhang, D., Ren, Y., Fan, X., Zhai, J. & Jiang, L. Photoassisted salt-concentration-biased electricity generation using cation-selective porphyrin-based nanochannels membrane. *Nano Energy* **76**, 105086 (2020).

Responses to Reviewer #2:

In this work, Tang et al. developed a kind of porous membrane with natural materials for osmotic energy conversion. With an interlocking configuration between the montmorillonite matrix and the intercalated cellulose nanofibers, the fabricated porous membrane presents strong mechanical strength and tunable widths of interlayer nanochannels. With the confined nanochannels and negative surface charges, the membrane has strong cation selectivity which is a good candidate for the high-performance osmotic energy conversion. The authors conducted MD simulations to provide the physical details of ionic diffusion across the porous membranes. Detailed life cycle assessment was also done to show the economic, environmental, and resource impacts of the developed membrane.

The manuscript can be improved by solving the following questions.

Response: Thank you for the careful review and helpful comments on our manuscript. We have revised our manuscript carefully according to your suggestions. We hope our revised version could satisfy the standards of this journal.

1. The manuscript was not prepared carefully. Many grammatical errors in the text should be revised carefully, such as:

Line 39

a salinity gradient

Line 140

'were' should be 'was'

Line 269

gradients

Line 337

an important factor

In the whole manuscript, 'pH' should be used correctly.

There are also some poor statements, like:

Line 293-295

The experimental conditions should be provided. Or it's hard to follow the description.

Line 421 'According to the literature reports'

While, there are no references provided in the sentence.

Response: Thanks for the very careful review and providing many helpful comments. We have checked the article carefully and revised some grammar and handwriting errors in the manuscript (**Page 2, 3, 4, 6, 8, 10, 11, 12, 20**).

Furthermore, we have optimized some poor statements to improve the quality of the manuscript according to your suggestions. For instance, we have provided the detailed experimental conditions for a better understanding. The modified contents are shown as follows:

(1) *Original description:* When the same salinity gradient was applied to KCl solution, the power density was up to 2.82, 10.38 and 17.06 W m⁻², respectively (Fig. S20), which mainly attributed to different ionic diffusion coefficients ($K^+ > Na^+$). (**Line 293-295 in original manuscript**)

Revised description: For a fair comparison, the same salinity gradients were applied to the KCl aqueous solution system, that is the high-salinity solution was fixed at 0.5 M KCl and the low-salinity one was controlled as 1, 10 and 100 mM. In this case, the obtained osmotic power density was respectively 2.82 (5-fold), 10.38 (50-fold) and 17.06 W m⁻² (500-fold) as shown in Fig. S25. A higher osmotic energy output in KCl system compared with NaCl system was mainly attributed to the higher ionic diffusion coefficient of K⁺ ion ($1.960 \times 10^{-9} \text{ m}^2 \text{ s}^{-1}$) than Na⁺ ion ($1.334 \times 10^{-9} \text{ m}^2 \text{ s}^{-1}$). (**Line 372-378, page 11 in revised manuscript**)

(2) *Original description:* Ultimately, according to the literature reports, 10% nanofiber composites were chosen to assess the impact on resources, environment and economy during the production for mainstream 2D material-based nanofluidics. (**Line 420-423 in original manuscript**)

Revised description: Detailed production processes of 2D materials and nanocellulose used for nanofluidics preparations in reported works (*Nat. Commun.*, 2019, 10, 2920; *Energy Environ. Sci.*, 2021, 14, 4400; *Mater. Horiz.*, 2020, 7, 2702; *ACS Cent. Sci.*, 2021, 7, 1486-1492; *Nano Energy*, 2022, 100, 107526) were summarized to evaluate the resource, environment and economic impacts of 2D-NNF developed in this work, which included the steps from raw material processing to laboratory exfoliation, and analyzed the material flow in detail (Fig. 6a,b and Table S7). **(Line 491-496, page 15 in revised manuscript)**

For the lack of references to support the description, we have added some helpful references in corresponding context of the revised manuscript. **(Line 493, page 15 in revised manuscript)**

(3) *Original description:* The velocity ratio (V_{K^+}/V_{Cl^-}) between K^+ and Cl^- ions in the vertical direction was 10 times larger than V_{K^+}/V_{Cl^-} in horizontal direction. **(Line 234-236 in original manuscript)**

Revised description: The velocity ratio (V_{K^+}/V_{Cl^-}) between K^+ and Cl^- ions in the horizontal direction was 1/10 of that V_{K^+}/V_{Cl^-} in vertical direction. **(Line 306-308, page 9 in revised manuscript)**

(4) *Original description:* Regarding nanocellulose, the environmental impact of CNF was found to be 10 times lower than that of ANF at an equivalent mass (Table S6), despite the involvement of a small amount of chemical reagents in the preparation process. **(Line 417-420 in original manuscript)**

Revised description: Regarding the nanocellulose, the environmental impact of CNF was 1/10 of that of ANF at an equivalent mass (Table S8), despite the involvement of a small amount of chemical reagents in the preparation process. **(Line 503-505, page 15 in revised manuscript)**

(5) *Original description:* Consequently, the energy consumption and abiotic depletion (ADP) in the production process of 2D-NNF is reduced by a factor of 10.2 and 13.9, respectively. Additionally, the corresponding reductions in GWP, AP, HTP, MATEP, and FAETP are 9.2, 12.9, 60.9, 44.5, and 14.0 times, respectively, in

comparison to the GO and MXene nanofiber-based membrane (Table S8). **(Line 440-444 in original manuscript)**

Revised description: Consequently, the energy consumption and abiotic depletion (ADP fossil) in the production process of 2D-NNF are only 2/15 and 1/139 of GO and MXene nanofiber-based membrane consumption, respectively. Additionally, the corresponding impact in GWP, AP, HTP, MATEP, and FAETP are 1/9, 1/13, 1/61, 1/45 and 1/14, respectively (Table S10). **(Line 522-526, page 16 in revised manuscript)**

(6) *Original description:* It can be seen from the current and power output curves that the 2D-NNF exhibited good stability in an acid and neutral condition (Fig. S21a). When the pH was increased to 11.00, the power density increased by about 13%. This could be associated with more counterions inside channels and increased negative surface charge due to deprotonation of surface functional groups. **(Line 349-353 in original manuscript)**

Revised description: It can be seen from the current and power output curves increase with the increase of pH (Fig. S26a), which could be attributed to the deprotonation of carboxyl groups on the surface of CNF. As a result, the corresponding output power density reached the maximum ($\sim 13.80 \text{ W m}^{-2}$) at pH=11 (Fig. S26b). **(Line 419-423, page 13 in revised manuscript)**

2. Line 28, Line 99, and Line 488. In this work, the authors prepared membranes with a diameter of 30 cm. However, the original membrane was not used directly for the osmotic energy conversion. The description in the three places should be changed to eliminate misunderstandings about osmotic energy conversion with the 30-cm-diameter membrane.

Response: Thank you for the helpful comments. To eliminate misunderstandings, we have revised the description in these three places of the original manuscript as follows according to your suggestions.

(1) *Original description:* When the 2D nanofluidic membrane is scaled, it could also delivery a uniform high-power output of over 8.0 W m^{-2} at any test sites as well as a long-term stability for 30 days (**Line 28 of the original manuscript**).

Revised description: When the area of 2D nanofluidic is scaled up to 700 cm^2 , several test sites were selected from different regions of the membrane could also delivery high-power output of over 8.0 W m^{-2} as well as a long-term stability for 30 days. (**Line 28 of the revised manuscript**).

(2) *Original description:* For the large-scale membrane with an area of 700 cm^2 , the average maximum power still reaches 8.36 W m^{-2} and could maintain long-term stability over 30 days, attributing to excellent uniformity and stability in both physical and chemical structures. (**Line 99 of the original manuscript**)

Revised description: For the large-scale membrane with an area of 700 cm^2 , the osmotic energy conversion was carried out by selecting several test sites from the different regions of membrane under the same test conditions. The average maximum power of different test sites still reaches 8.36 W m^{-2} and could maintain long-term stability over 30 days, attributing to excellent uniformity and stability in both physical and chemical structures. (**Line 102-107, page 3 in the revised manuscript**)

(3) *Original description:* When the area of the 2D-NNF is enlarged to 700 cm^2 , it still exhibited high power density generation ($\sim 8.33 \text{ W m}^{-2}$) and long-term stability (>30 days). (**Line 488 of the original manuscript**).

Revised description: When the area of the 2D-NNF is enlarged to 700 cm^2 , the different sites selected from various regions of membrane could still exhibit high power density generation ($\sim 8.36 \text{ W m}^{-2}$) and long-term stability (>30 days) (**Line 571-577, page 18 in the revised manuscript**).

3. During the fabrication of the porous membrane, are there any experimental procedures that ensure the roughly uniform film thickness/ porosity over a larger area? How many membranes had been prepared in this work? Does the thickness (or other physical parameters) demonstrate good repeatability?

Response: Thank you for these helpful comments. During the manufacturing process of porous membranes, there are mainly four experimental procedures to ensure the roughly uniform film thickness/porosity over a larger area as follows.

(1) Preparing a uniform dispersion as the precursor solution for vacuum filtration. A good uniformity and dispersion of precursor is a key to prepare a membrane with uniform thickness and porosity. The uniform precursor solution was obtained by a vigorous stirring and ultrasonic dispersion, followed by collecting the supernatant after precipitation for 24 h. Then, the thickness of the membrane could be well regulated by controlling the volume of precursor solution.

(2) Conducting a suitable pressure regulation during the vacuum filtration. The equipment used to prepare the large-area porous membrane was a positive-pressure filter device as illustrated in **Fig. R1a**, rather than the conventional negative-pressure vacuum filtration ones. This could achieve an accurate control of pressure and ensure the uniform consistency of pressure everywhere in the top-down self-assembly process of the membrane. In this device, the grid-like filter substrate is used to prepare a flat porous membrane when conducting a suitable pressure. A higher pressure will affect the flatness and uniformity of the large-area membrane because of the mechanical deformation (**Fig. R1b**).

(3) Selecting an appropriate filter membrane. In this work, the cellulose acetate filter membranes were used for the preparation of 2D-NNF because of its proper pore size ($\sim 0.22 \mu\text{m}$) and excellent hydrophilicity. Furthermore, a polypropylene filter membrane was employed as the hard support layer to avoid the grid imprinting (from the grid-like filter substrate) formed on the membrane surface (**Fig. R1c**) and caused the uneven thickness.

(4) Controlling the proper drying conditions. The drying temperature and duration will affect the quality of the large-area membrane. For instance, too long drying duration or too high temperature will make the filter membrane deformation and thus affect the flatness of the 2D-NNF membrane (**Fig. R1d**), which in turn lead to a poor uniform of the film thickness.

Fig. R1 (a) The positive-pressure filter device used for large-area membrane preparation. (b) The membranes obtained under different pressures. (c) The obtained membrane without usage of hard support membrane had the grid imprinting (from the grid-like filter substrate) on the surface. (d) The effect of drying duration on the uniformity of the large-area membrane.

In order to obtain a reliable performance, we have prepared dozens of membranes and about ten high-quality ones were used for the detailed study in the manuscript. To evaluate the reversibility of the scaled-up membrane, three membranes prepared using the same process were selected randomly for the investigation of two critical physical parameters that effect the ion transport behaviors, that is the membrane thickness and channel size (interlayer spacing). For a comparison, nine different regions were selected from each membrane for the following characterizations (**Fig. R2, left**). The uniformity and accurate thicknesses of the membranes were determined by the cross-sectional SEM images. As shown in **Fig. R2 (right)**, all the membranes have the uniform

thicknesses in different regions (**Fig. R3a**), and the thickness of three membranes were almost the same when using the same preparation conditions (**Fig. R3b**). This indicates that the thickness of the membrane has good consistency and reversibility.

Fig. R2 Photographs and cross-sectional SEM images of selected regions from large-area 2D-NNF films with the same preparation conditions. (a) $15.97 \pm 0.43 \mu\text{m}$; (b) $15.32 \pm 0.48 \mu\text{m}$; (c) $16.00 \pm 0.56 \mu\text{m}$. Scale bar: $20 \mu\text{m}$.

Fig. R3 The thickness of different sites from different membranes with the same preparation conditions. (b) Average thickness of different membranes.

Then, the interlayer spacing (channel size) was studied based on the above three large-area membranes. As shown in **Fig. R4**, there is little shift in the Small-angle XRD patterns for different sites of different membranes. The calculated interlayer spacing of the three membranes are respectively 1.25 ± 0.02 nm, 1.24 ± 0.03 nm and 1.23 ± 0.03 nm (**Fig. R4d**), indicating good consistency and reversibility of the critical physical parameters of large-area membranes.

For a better understanding, we have added above discussions in our revised manuscript (**Page 14, Fig. S28-30**).

Fig. R4 Small-angle XRD patterns of the different sites of large-area membranes with the same preparation conditions.

Furthermore, the thickness of the large-area membrane could be flexibly regulated as shown in **Fig. R5**. The membrane with different thickness also demonstrated good thickness uniformity in different regions. The average error control of different membrane thickness was about $\pm 0.5 \mu\text{m}$ (**Fig. R6**).

Fig. R5 Photographs and cross-sectional SEM images of selected regions from large-area 2D-NNF films of different thicknesses (a) $6.36 \pm 0.38 \mu\text{m}$; (b) $15.32 \pm 0.48 \mu\text{m}$; (c) $19.18 \pm 0.41 \mu\text{m}$. Scale bar: $20 \mu\text{m}$.

Fig. R6 The thickness of different sites of the samples with different thickness. (b) Average thickness of different samples.

4. The nanochannel shows expansion in the fully hydrated state. Does the expansion depend on the applied solutions which have different Debye lengths?

Response: Thank you for this insightful comment. The Debye length (λ_D) could be calculated according to the following equation (Eq. R1),

$$\lambda_D = \sqrt{\frac{\varepsilon_0 \varepsilon_r K_B T}{e^2 \sum c_i z_i^2}} \quad (\text{R1})$$

where ε_r and ε_0 are respectively relative dielectric constant (~ 78.5 at room temperature) and vacuum permittivity ($\sim 8.854 \times 10^{-12}$ F/m), K_B is the Boltzmann constant (1.38×10^{-23} m² kg s⁻² K⁻¹), T and e are respectively temperature (298 K) and elementary charge (1.6×10^{-19} C), c_i and z_i are concentration and valence number of ions in the solution. For the solution containing same ions, the λ_D only depends the solution concentration. A higher concentration leads to a shorter λ_D according to Eq. R1. Therefore, a series of KCl solutions with different concentrations were applied to clarify the influence of λ_D on the expansion of nanochannels in the 2D-NNF. The nanochannel size (interlayer spacing) was measured with small-angle XRD. Before the measurement, the 2D-NNF membrane was immersed in KCl solution with different concentration for 24 h and then wiped the residual solution on the surface of membrane. As shown in **Fig. R7**, the nanochannel size increases slightly with the increasing λ_D due to the enhanced thickness of double electrode layer (equal to λ_D). Consequently, the expansion behavior of nanochannels depend on the applied solutions with different Debye lengths.

For a better understanding, we have added the follow figure and corresponding discussions in our revised manuscript (**Page 7, Fig. S12**).

Fig. R7 Evolution of nanochannel size (interlayer spacing) of the hydrated 2D-NNF in the applied solutions with different Debye lengths.

5. The authors claimed that the surface charges originate from the deprotonation of carboxyl and hydroxyl groups. (Line 117) During the preparation of the membrane, is the aqueous solution the same in concentration and pH? As shown in Figure S7, why does the surface charge density increase with the salt concentration? Under 1 M KCl, the surface charge density can reach ~ -65.6 mC/m². Also, why does the surface charge density have no dependence on the solution pH? Is there any consideration of the concentration of H⁺ and OH⁻ when using equation S2, especially at a very large or small pH?

Response: Thanks for these very helpful comments. According to these comments, we have carried out an in-depth study on the surface charge evolution of 2D-NNF with the varied salt concentration and pH value of the solution. This helps to obtain more convincing experimental results and deepen our understanding of the important concept of nanofluidics, which is of great value in improving our further works.

For the first question, the aqueous solution used for the preparation of membrane was the deionized water (18.2 MΩ cm) without pH value modulation. Thus, the concentration and pH value (~ 5.7) of the solution were almost unchanged during the

preparation of the membrane. To demonstrate the surface charge property of the membrane in various operating environments, a series of aqueous solutions with different concentrations and pH values were applied for the following zeta potential measurements and surface charge calculations.

For the question: “why does the surface charge density increase with the salt concentration?” In aqueous solution, the generation of surface charge on the 2D-NNF is derived from the deprotonation of carboxyl and hydroxyl groups, which could adsorb counterions to form an electric double layer (EDL) that has the same thickness with the Debye length (λ_D). The existence of EDL will produce a zeta potential (ζ), from which we can calculate the surface charge density (σ) according to following equation (Eq. R2),

$$\sigma = \frac{\varepsilon_0 \varepsilon_r \zeta}{\lambda_D} \quad (\text{R2})$$

where ε_r and ε_0 are respectively relative dielectric constant and vacuum permittivity, ζ is the zeta potential, and λ_D is the Debye length that could be calculated according to Eq. R1. Therefore, the σ value is determined by both of measured ζ and calculated λ_D . As shown in **Table R2**, the measured ζ value has little difference with varied concentration, while the calculated λ_D value obviously decreases with the increasing concentration. Consequently, the surface charge density increases with the salt concentration according to Eq. R2.

Table R2 The measured ζ and calculated λ_D corresponding to solutions with different concentrations.

KCl concentration (mM)	Measured ζ (mV)	Calculated λ_D (nm)
0.1	-33.2	30.5
1	-31.9	9.6
10	-32.6	3.1
100	-38.7	1.0
1000	-29.8	0.3

For the question concerning dependence of surface charge on the solution pH, we summarized the influence of solution pH on the surface charge in two parts. One is the determination of positive or negative **polarity of surface charge**, which is ascribed to the protonation or deprotonation across the isoelectric point of surface groups. The other is the determination of **magnitude of surface charge** (charge density), which is attributed to the substantial change in solution concentration when introducing additional ions (H^+ or OH^-) at high or low pH. We are sorry that the latter was neglected in our original manuscript. Thus, the surface charge variation with solution pH has been restudied considering the concentration change with pH value based on the reminder of the reviewer.

Surface charge polarity influenced by solution pH: The 2D-NNF is composed of montmorillonite (MMT) and carboxylated cellulose nanofibers (CNF). The surface charge of MMT crystal includes the excess permanent charge (90%) and few variable charge (10%) (*Appl Clay Sci*, 2004, 27, 75-94; *J. Colloid Interface Sci.* 2013, 390, 225-233). The permanent charge is generated by isomorphic substitution of MMT crystal. The structural unit of MMT is a layer of aluminum oxide octahedron sandwiched between two layers of silica tetrahedron. The substitution of high valence ions (Si^{4+} , Al^{3+}) by low valence ions (Mg^{2+}) results in the formation of surface negative charge. The polarity and density of such charge could not be affected by the solution conditions (*Langmuir*, 2013, 29, 14926-14934), so called the permanent charge. The variable charge formed on the surface of MMT is derive from the protonation or deprotonation of surface hydroxyl groups (-OH), which could be modulated by the solution conditions especially the pH value. The isoelectric point of MMT is about 9, thus the charge polarity of variable charge changes with the varied pH value across 9. In the MMT, the permanent negative charges are 90 percent of the total charge, which determines the charge polarity and density of the materials. As a result, the zeta potential of MMT remained stable with varied solution pH (**Fig. R8, blue line**). For the CNF, the surface charge is generated from the protonation or deprotonation of excess carboxyl groups ($pK_a \sim 3.8$), which thus carries negative charges in the pH range of our measurement

(3~11). Therefore, the introduction of CNF obviously enhances the negativity of zeta potential of MMT, and the value varied with solution pH (**Fig. R8, red line**).

Fig. R8 Zeta potential of pristine MMT and 2D-NNF (CNF intercalated MMT) as a function of pH value of 10 mM KCl solution.

Surface charge density influenced by solution pH: According to our previous discussions, the surface charge density of 2D-NNF is highly dependent on the ion concentration of solution. In this case, modulation of solution pH will inevitably introduce excess H^+ or OH^- ions and thus increase the ion concentration of solution, especially for low-concentration and extreme pH values. Therefore, we have recalculated the surface charge density in response to solution pH value considering the concentration increase of extra H^+ or OH^- ions. As shown in **Fig. R9**, the surface charge density is higher at the two extreme pH (3, 11) because a higher ion concentration shortens the λ_D value. For the medium pH value (5, 7, 9), the concentration variation caused by few extra ions is almost negligible, and thus the charge density increases with pH value as that of zeta potential.

Fig. R9 Surface charge density and zeta potential of 2D-NNF as a function of pH value of 10 mM KCl solution.

Based on above discussions, we have corrected the thoughtless descriptions and figures about the pH-dependent surface charge in our revised manuscript (**Page 5, Fig. S9**).

6. The hopping mechanism is usually used to describe the transport of protons. The authors need to provide more statements about the applicability of the hopping mechanism in the diffusion process of K ions. In aqueous solutions, K ions are hydrated and much larger than protons. As shown by the inset of Fig.2a, is the ionic hopping along the horizontal direction in Fig.2f? However, the MD simulations show similar velocities of K and Cl ions. More discussions can be provided.

Response: Thank you for these valuable comments. We have carried out detailed calculations to give an in-depth understanding of ion transport behaviors in the 2D nanofluidics.

For the question concerning hopping mechanism in the diffusion process of K ions.

In aqueous environment, the cluster of hydrated K^+ with water molecules indeed has a much larger diameter (6.72 Å) than that of proton, considering additional K-O coordinate bond. The K ions undergo a dehydration process before entering the nanochannel, while the ionic radius of K^+ following dehydration is 1.33 Å. Moreover, DFT calculations indicated the hydration energy of K^+ and Cl^- are 3.07 eV and 3.72 eV,

respectively, see **Fig. R12**, while the Cl^- hydration is based H-Cl hydrogen bond. Therefore, K^+ is easier to enter the nanochannel compared with Cl^- , given low hydration energy.

On the other hand, in order to understand the hopping mechanism of K^+ diffusion process, the simulations of K^+ transport in 2D nanofluidic membrane were performed via Ab initio molecular dynamics (AIMD). The hopping mechanism was initially proposed to explain the ion transport behavior when the double electric layer (EDL) is fully covered by the ions. Diffusion EDL was mainly derived from Gouy-Chapman-Stern model (*Phys. Fluids.*, 2005, 17, 100604), which consists of the Stern layer and diffuse layer. While, the diffuse layer was composed of K^+ in our modeling two-dimension all-natural nanofluidic (2D-NNF)/ K^+ system. The 2D-NNF was coated with a layer of K^+ , and each K^+ has interactions with the O at the surface of 2D-NNF. At the beginning of MD simulations, the additional K^+ with velocity collided with one K^+ at rest in the diffuse layer, and the other K^+ in the diffuse layer moved with pendulum-like motion, along the additional K^+ velocity direction. K^+ far away from the collision moved 1.5 nm following 80 ps in the diffuse layer, through pendulum-like motion, see **Fig. R10**. The K^+ transport in MMT presented energy transfer hopping process. According to your suggestion, we have added the above discussion in the revised version (**Page 7, Fig. S13**).

Fig. R10 Ab initio molecular dynamics simulations of K^+ transport in diffuse layer.

For the question: “As shown by the inset of Fig.2a, is the ionic hopping along the horizontal direction in Fig.2f? However, the MD simulations show similar velocities of K and Cl ions. More discussions can be provided.” The concentration of KCl solution in MD simulations was set as 1 mol/L (or 10^3 mM). If we keep the same numbers of both K^+ and Cl^- ions as those of MD simulations in this work, the volume and atom number of modeling system need to extend for 10^4 times, for instance with low KCl concentration of 10^{-1} mM for hopping-ion transport study in 2D-NNF. It will be too time-consuming to perform. We therefore investigated the high concentration for only bulk-like ion transport with MD simulations in Fig. 2f, rather than the hopping-ion transport as inset of Fig. 2a.

The difference in the transport velocity between vertical and horizontal direction was observed for bulk-like ion transport, which attributed to the electrostatic and dehydration of ions when entering the vertical channel. DFT calculations implied the hydration energy of K^+ is approximately 3.07 eV, while that of Cl^- is about 3.72 eV. Consequently, K^+ undergoes the dehydration process more easily than Cl^- when entering the channel. Once inside the horizontal channel, the ions complete the dehydration process, and the electrostatic forces on both sides of the nanochannel are symmetric. Therefore, MD simulations showed similar velocities for both K^+ and Cl^- ions along the horizontal direction. According to your suggestion, we have added the above discussion in the revised version (**Page 9-10, Fig. S19**). Details of DFT theoretical calculations were as follows, “From the calculation results of MD, the water molecules near K^+ ion and Cl^- ion were obtained with DFT calculation at the level of M06-2X/def2-TZVPP, while the hydration were exported from MD equilibrium structure. It implied the hydration energy of K^+ is 3.07 eV, while that of Cl^- is 3.72 eV. Therefore, K^+ undergoes the dehydration process more easily than Cl^- when entering the 2D-NNF.” (**SI Page 4**)

Fig. R11 Configurations for the hydration of K^+ ion (purple) and Cl^- ion (green) exported from MD calculations.

7. The nanochannels in the membranes have sub-2 nm widths (Fig.1e). As shown in Fig S1b, the nanofibers have an averaged diameter of ~6 nm, much larger than 2 nm. Then, how can the nanofibers tune the surface charge density of MMT channels? A scheme may be needed to show the structure of the membrane. The microscale structure of the membrane is important which determines the conduction of MD simulations and the ionic behaviors.

Response: Thanks for the very helpful comments. The nanochannel size of the 2D membrane was determined by the value of interlayer spacing, which is characterized by the small angle X-ray diffraction (SAXRD). This characterization is a feedback to the parallel modulation interface with good periodicity in the lamellar membrane. As a result, the interlayer spacing obtained by the SAXRD is an average size of a periodic parallel arrangement. In a real experiment, few nanofibers were randomly intercalated into the lamellar membrane in the preparation of 2D-NNF, leading to a random expansion in some spaces of the nanochannels as illustrated in Fig. R12. Although few nanochannels could be enlarged by nanofibers, these nonperiodic regions cannot be reflected from the SAXRD (*ChemSusChem*, 2022, e202200933). Therefore, the measured interlayer spacing (sub-2 nm) is smaller than the diameter (6 nm) of unconfined nanofibers.

Fig. R12 Schematic diagram for the microstructure of the 2D-NNF with periodic interlayer nanochannels and random enlarged nanochannels with CNFs intercalation.

The nanofibers (CNFs) carry abundant surface negative charges as discussed above, which could enhance the local space charge density in the enlarged nanochannels (**Fig. R12**). This result has also been obtained in some previously reported works (*Nat. Commun.*, 2019, 10, 2920; *Energy Environ. Sci.*, 2021, 14, 4400–4409; *J. Am. Chem. Soc.* 2021, 143, 1932). The space charge of CNFs could enlarge the EDL region in the nanochannels, contributing to higher cation selectivity and faster cation transport (*Nat. Commun.*, 2019, 10, 2920; *J. Am. Chem. Soc.*, 2022, 144, 13764-13772; *Nano Energy*, 2022, 92, 106709). Consequently, the MMT membrane with CNFs intercalation exhibits a faster ion transport dynamics and a superior osmotic energy output compared with pristine MMT membrane. Furthermore, the MD simulations were also conducted based on the model with higher charge density considering the charge regulation of CNFs, which is consistent with the experimental results.

For a better understanding, we have added above scheme and corresponding discussions in our revised manuscript (**Page 5, Fig. S8**) to show the microstructure of the 2D membrane.

8. More details about the MD simulations should be provided, like how many water molecules, cations, anions, and charges in the simulation systems.

Response: Thanks for the helpful comment. We have added those details in molecular dynamics simulations in revised manuscript (**Page 8**). The modeling system of $5.2 \times 13.5 \times 30.0 \text{ nm}^3$ was built, see **Fig. R13**, which have a total of 197975 atoms for 2D-NNF/KCl/H₂O configuration, and the whole system is electrically neutral. While, there are 1086 K⁺ cations and 1086 Cl⁻ anions, which are compose of 1 mol/L KCl solution with 58161 water molecules. The modeling 2D-NNF was constructed with the layer distance of 1.90 nm (Fig. S7b), which was obtained from XRD patterns. The equilibrium structure was obtained, following at 12 ns MD simulations. The negative charges surface were induced by adjusting the Mg:Al atomic ratio in the 2D-NNF structure (Supplementary Note 7). The surface charges of $57 \text{ mC} \cdot \text{m}^{-2}$ was obtained, which was derived from the ratio of the total charges of ions on the 2D-NNF surface in MD equilibrium structure, and it was consistent with the measured $66 \text{ mC} \cdot \text{m}^{-2}$ in experiment (Fig. S9b).

Figure. R13 The parameters in the MD simulations.

9. 12 ns may be not long enough to obtain analyzable data. From Fig. 2e, the number of cations diffusing through the membrane is only ~10. The authors can explain why the diffused ion number shows a large decrease at ~6 ns.

Response: Thanks for the helpful comments. The K^+ , Cl^- ions as well as water molecules were randomly inserted in the modeling system, therefore, the equilibrium structure was not achieved in initial MD simulations ($t=0$ ns). For instance, the modeling solution was not fully mixed, as well as, the concentration and the diffusion through the membrane were not balanced. That is the reason, as reviewer mentioned, “diffused ion number shows a large decrease at ~6 ns”. When the system reaches a state of thermodynamic equilibrium in MD calculations, there were not significant variation in all physical parameters as a function of time. Therefore, we updated the MD result up to 20 ns in revised version, rather than previous 15 ns in last version. For instance, the diffused ions (both K^+ and Cl^-) number only have minimal change from $t=12$ ns to $t=20$ ns in MD result, see following **Fig. R14**. For the convenience of readers, we marked the equilibrium and non-equilibrium regions in Fig. 2e of **revised manuscript (Page 9)** as well, and added the following text, “when the equilibrium structure was achieved following 12 ns MD simulations, the diffused ion numbers through 2D-NNF were hardly changed anymore, and the migration ratio between K^+ and Cl^- ions in 2D-NNF was exported, which is 2.36 (Fig. 2e).”

Figure. R14 The number of ions that passed through the nanochannels in the MD simulation as a function of time.

10. For equation S5, the ion activity coefficients should be provided for the solutions used in the work.

Response: Thank you for this helpful comment. The different solutions used in the work were KCl, NaCl, LiCl, CaCl₂ and MgCl₂ aqueous solution with different concentration. The ion activity coefficients of dilute solutions could be calculated using the simplified Debye-Hückel equation (Eq. R3),

$$\lg\gamma_{\pm} = -A |z_+z_-| \sqrt{I} \quad (\text{R3})$$

where γ_{\pm} is ion average activity coefficient, 'I' is the ionic strength, z is ion valence number, and A is constant, the value is about 0.509 at 298 K. The calculated ion activity coefficients are summarized in the following table (**Table R3**).

Table R3 Average activity coefficients of various aqueous solutions with different ions and concentrations at 298 K.

Concentration/M	0.001	0.01	0.05	0.1	0.5	1
KCl	0.965	0.901	0.815	0.769	0.650	0.605
NaCl	0.966	0.904	0.823	0.778	0.682	0.658
LiCl	0.964	0.889	0.769	0.790	0.739	0.774
CaCl ₂	0.887	0.724	0.574	0.518	0.448	0.500
MgCl ₂	0.879	0.816	0.635	0.528	0.480	0.569

For a better understanding, we have added above table and corresponding discussions in the revised manuscript (**SI Page 3, Table S2**).

11. For the natural situation at the estuaries, the concentration of the seawater is ~0.5 M. The sentence ‘2D-NNF could still work in high salt solution’ makes me puzzled.

Response: Sorry for the unclear description. The actual mean of this sentence is that “2D-NNF could exhibit a higher performance in high salinity-gradient solutions.” In this work, we studied the osmotic energy harvesting under different salinity-gradient by fixing high salt solution at 0.5 M and varying low salt solution from 0.001 to 0.1 M. As a result, a maximum output power density of 15.87 W m^{-2} was obtained at 500-fold salinity gradient. For a better understanding, we have revised the description of this sentence in our revised manuscript (**Page 10-11**) as follows.

“Along with the concentration gradient increasing, the produced current density gradually increased, and the output power density reached up to 15.87 W m^{-2} at 500-fold salinity gradient, demonstrating the 2D-NNF could be adapted to different river water concentration conditions.”

Furthermore, some high salt solutions that simulate other water environments were constructed to study the performance of 2D-NNF under high polarization conditions. The artificial seawater was replaced by artificial brackish water (0.2 M), desalination brine (1.3 M), mining waste water (3.0 M) and salt-lake water (4.3 M) as shown in **Fig. R15**. A maximum power density of 60.46 W m^{-2} could be achieved from the 2D-NNF based osmotic energy harvesting system at the salt lake/river condition. Furthermore, the diffusion current could maintain a continuous operation for a long-term measurement without electrolyte replenishing. For a better understanding, we have added above discussions in our revised manuscript (**Page 11, Fig. S21**).

Fig. R15 (a) The current densities and (b) power densities at different salinity conditions as a function of the external resistance. Power generation and conversion efficiency (c) and long-term current output (d) under a series of artificial water resources including brackish water, desalination brines, brines from mining activities, and water from salt-lake. Error bars represent S.D.

12. At pH 11, the surface potential seems to remain the same from Fig. S7. The pKa value for carboxyl groups is ~3.8, in solutions with pH higher than 8 or 9, all groups should be deprotonated. The surface charge density can reach its largest value. Maybe more discussion can be added here.

Response: Thanks for the careful review and helpful comments. We have revised the performance figure and discussions about the pH-dependent of surface charge density as discussed in the response of comment 5. More discussions have been added into our revised manuscript (**Page 5**) as follows.

“The surface charge density of 2D-NNF is highly dependent on the ion concentration of solution. In this case, modulation of solution pH will inevitably introduce excess H⁺ or OH⁻ ions and thus increase the ion concentration of solution, especially for low-concentration and extreme pH values. Therefore, the surface charge density evolution with varied solution pH was calculated considering the concentration increase of extra H⁺ or OH⁻ ions. The results show that the surface charge density is higher at the two extreme pH (3, 11) because a higher ion concentration shortens the λ_D value. For the medium pH value (5, 7, 9), the concentration variation caused by few extra ions is almost negligible, and thus the charge density increases with pH value as that of zeta potential.”

13. From the nanofluidic experiments, the fabricated membrane exhibits high-performance osmotic energy conversion. Is it possible to conduct the osmotic energy generation experiment with the 30-cm-diameter membrane directly? If so, this is much closer to the real application.

Response: Thank you very much for the constructive comments that help to improve our further works. We have designed the special equipment for direct studying the osmotic energy output of the large-area membrane (φ 30 cm) as illustrated in **Fig. R16**. The self-standing membrane could be well immobilized in equipment for the following measurements (**Fig. R16a**). After injecting solutions on both sides, the membrane could still maintain a good mechanical property (**Fig. R16b**), providing a good guarantee for long-term measurements.

Fig. R16 Special equipment for osmotic energy harvesting study of large-area 2D-NNF membrane (φ 30 cm).

Then, the osmotic energy harvesting behaviors were studied in the simulated seawater (0.5 M NaCl) /river condition (0.01 M NaCl). The effective test area was φ 20 cm, and two Ag/AgCl electrodes were applied for the measurement. As shown in **Fig. R17a**, the obtained V_{OC} was about 0.12 V, which is close to the small-area membrane, while the I_{SC} (3.37 mA) was significantly enhance compared with small-area membrane (several micro ampere) owing to the enlarged permeability area. Furthermore, the current output was stable for a long term over 120 min, when the current retention reached 95% of the maximum value (**Fig. R17b**). The results show that the 30-cm-diameter membrane could be directly used for osmotic energy harvesting and output

considerable current and voltage, indicating great potentials in real-world applications.

However, when connecting external load, the output power of the large-area membrane ($\sim 70 \mu\text{W}$) was higher than that of the small-area ones ($\sim 0.3 \mu\text{W}$), while the calculated power density is much lower owing to the out of proportion between output power and test area. This is a common problem of nanofluidic systems that the power density increasing with the test area. Thus, it is still a big challenge to achieve comparable power output for the scale-up of the nanofluidic membranes, which is also important research directions of our further studies. For a better understanding, we have added above discussions in our revised manuscript (**Page 14, Fig. S37 and S38**).

Fig. R17 (a) Typical I - V curve of the large-area 2D-NNF with a diameter of 30 cm under the condition of simulated seawater/river salinity gradient (0.5 M/0.01 M NaCl). (b) Long-term current output of the large-area 2D-NNF membrane.

Responses to Reviewer #3:

The authors developed a highly efficient osmotic energy generator using natural clay-based nanofluidics. They created a two-dimensional nanofluidic system by combining montmorillonite nanosheets from clay and cellulose nanofibers from wood, enabling rapid cation transport and achieving an impressive osmotic power output of 8.61 W m^{-2} , surpassing previous 2D nanofluidic systems. The membrane's scalability and stability were also demonstrated, making it suitable for practical osmotic energy harvesting. I found the study intriguing and believe it enhances our understanding of osmotic energy production. However, there are concerns that the authors need to comprehensively address.

While their paper shows promise, a thorough revision is necessary before it can be accepted for publication.

Response: Thank you for the careful review and helpful comments concerning our manuscript. We have revised our manuscript carefully and thoroughly according to your suggestions. We hope our revised version could satisfy the standards of this journal.

1. The graphical abstract is one of the main pillars of the article, and at the same time, it should include the main topic. Authors need to include more details about nanochannels in the graphical abstract to make it more informative.

Response: Thanks for your helpful comment. we have added more details about the microstructure and ion transport behaviors of nanochannels in the graphical abstract (bottom right) as illustrated in **Fig. R18**. The supplemental enlarged microstructure demonstrates that the 2D-NNF has horizontal and vertical nanochannels for ions transport. The further enlargement shows the negatively-charged MMT and nanofibers both contribute to a selective and fast transport of cations, which is the key to achieve high-performance osmotic energy harvesting. For a better understanding, we have added the updated graphical abstract and corresponding discussions in our revised manuscript (**Page 3-4, Fig. 1a**).

Fig. R18 Schematic diagram for assembly of natural montmorillonite (MMT) and natural cellulose nanofibers (CNFs) into a large-area 2D-NNF membrane with selective and fast cation transport nanochannels.

2. It is suggested to give further details of the findings in the abstract.

Response: Thanks for your helpful comment. For a better understanding, we added further details of the findings in the abstract of revised manuscript (**Page 1**) as follows.

“Two-dimension nanofluidics constructed based on naturally abundant clay are good candidates for harvesting blue osmotic energy between the sea and river from the perspective of commercialization and environmental sustainability. However, clay-based nanofluidic membranes outputting long-term considerable osmotic power remain extremely challenging to achieve due to the lacks of surface charge and mechanical strength. Here, a two-dimension all-natural nanofluidic (2D-NNF) is developed as robust and high-efficient osmotic energy generator based on an interlocking configuration of the stacked montmorillonite nanosheets (from natural clay) and their intercalated cellulose nanofibers (from natural wood). The generated nano-confined interlamellar channels with abundant surface and space negative charges facilitate selective and fast hopping transport of cations in the 2D-NNF. This contributes to a remarkable osmotic power output of $\sim 8.61 \text{ W m}^{-2}$ by mixing artificial seawater and river water, much higher than all reported state-of-the-art 2D nanofluidics. When the area of 2D nanofluidic is scaled up to 700 cm^2 , several test sites were selected from different regions of the membrane could also delivery high-power output of over 8.0 W

m⁻² as well as a long-term stability for 30 days. The excellent structure uniformity and stability of large-area 2D-NNF is the basic to achieve real-world applications in natural osmotic energy harvesting. According to detailed life cycle assessments (LCA), the 2D-NNF demonstrates great advantages in resource consumption (1/14), greenhouse gas emissions (1/9), and production costs (1/13) compared with the mainstream 2D nanofluidics, promising a good sustainability for large-scale and highly-efficient osmotic power generation.

3. The text does not mention specific electrokinetic phenomena in nanofluidic systems such as ionic current rectification, ionic concentration polarization, energy harvesting, and reduced fouling. To this end, the following relevant references must be cited properly in the introduction and throughout the manuscript:

- <https://doi.org/10.1039/D0CP05974A>
- <https://doi.org/10.1016/j.electacta.2021.139221>
- <https://doi.org/10.1039/D2CP01015A>
- <https://doi.org/10.1016/j.electacta.2022.141175>
- <https://doi.org/10.1021/acs.langmuir.2c01790>

Response: Thank you for the helpful comments and providing these useful references to improve our manuscript. We have studied these specific electrokinetic phenomena in nanofluidic systems carefully according to these helpful references, and find that ionic current rectification is a very important characteristic for reducing ionic concentration polarization of nanofluidics in osmotic energy harvesting system (*Phys. Chem. Chem. Phys.*, 2021, 23, 2211; *Electrochim. Acta*, 2021, 395, 139221; *Electrochim. Acta*, 2022, 431, 141175; *Langmuir*, 2022, 38, 10313–10330). This provides us important research directions for constructing 2D nanofluidics with ion rectification towards enhanced osmotic energy harvesting. For a better understanding, we have added the importance of ion rectification in nanofluidics towards high-performance osmotic energy harvesting in our revised manuscript (**Page 2**), and also cited these helpful references

in appropriate positions of the text. (Page 2, 13)

4. The leading cause of mass transfer in nanochannels is the formation and distortion of electrical double layers (EDLs). The text of the manuscript has almost nothing on this issue. Authors must adequately clarify EDL theories and the factors that influence them. To this end, the authors can cite:

- <https://doi.org/10.1016/j.electacta.2021.139376> ,
- <https://doi.org/10.1021/acs.analchem.2c04559> ,
- <https://doi.org/10.1063/5.0160678>

Response: Thanks for providing these helpful comments and useful references. We strongly agree with the reviewer that the formation and distortion of EDLs is the leading of ion transport in nanofluidics. To highlight the importance of EDL, we have added detailed EDL theories in our revised manuscript (Page 7), where these helpful references have been cited well. The detailed discussions about the formation and distortion of EDLs are shown as follows.

For the formation of EDL: In aqueous solution, the negatively-charge surface of 2D-NNF could adsorb counterions to form electric double layer (EDL) in the nanochannels. The EDL region plays a leading role in regulating ion transport to achieve cation selectivity and fast cation conduction (*Electrochim. Acta*, 2021, 399, 139376; *Phys. Fluids*, 2023, 35, 082006; *Anal. Chem.*, 2023, 95, 1522-1531). Consequently, the nanochannel with full-filled EDL demonstrates superior ion regulation behaviors and thus a high osmotic energy harvesting.

For the distortion of EDL: The thickness of EDL is equal to the value of Debye length (λ_D), which could be calculated according to the following equation (Eq. R4),

$$\lambda_D = \sqrt{\frac{\varepsilon_0 \varepsilon_r K_B T}{e^2 \sum c_i z_i^2}} \quad (\text{R4})$$

where ε_r and ε_0 are respectively relative dielectric constant (~ 80 at room temperature) and vacuum permittivity ($\sim 8.854 \times 10^{-12}$ F/m), K_B is the Boltzmann constant (1.38×10^{-23} m² kg s⁻² K⁻¹), T and e are respectively temperature (298 K) and elementary charge

(1.6×10^{-19} C), c_i and z_i are concentration and valence number of ions in the solution. For the solution containing same ions, the λ_D only depends the solution concentration. Thus, the varied solution concentration will cause the distortion of EDL, and a higher concentration leads to a thinner EDL (shorter λ_D) as demonstrated in **Table R4**. Therefore, small-sized nanochannels are desirable to achieve effective EDL occupation for reliable ion transport regulation and thus obtain high-efficient osmotic energy output.

Table R4. Distortion of EDL with the concentration of KCl aqueous solution.

KCl concentration (mM)	EDL thickness (nm)
0.1	30.5
1	9.6
10	3.1
100	1.0
1000	0.3

5. The authors should elaborate on the interactions between montmorillonite nanosheets and cellulose nanofibers at the molecular level. How do these interactions contribute to the structural stability and functionality of the membrane?

Response: Thank you for this helpful comment. At the molecular level, an interlocking configuration was formed based on the chemical bonding between montmorillonite (MMT) nanosheets and cellulose nanofibers (CNFs) as illustrated in **Fig. R19a**, which was confirmed by the X-ray photoelectron spectroscopy (XPS) and Fourier transform infrared (FTIR) spectroscopy characterizations. As shown in the XPS fine spectra of C1s (**Fig. R19b, c**), the appearance of C=O peak (288.4 eV) of 2D-NNF compared with pristine MMT confirmed the existence of CNFs. Also, there is a remarkable increase in the peak areas of C-O and C-C with the introduction of CNFs. In the fine spectra of Al 2p and Si 2p (**Fig. R19d, e**), the obvious chemical shifts for the characteristic peaks of 2D-NNF compared with pristine MMT were attributed to the formation of covalent bonds (Si-O-C and Al-O-C) between MMT and CNFs (*Science*, 2007, 318, 80-83, *ACS Appl. Mater. Interf.*, 2016, 8, 34914-34923). As shown in the FTIR spectra (**Fig. R20**), the introduction of CNFs contributed to the enhanced of hydroxyl (-OH) in 2D-NNF (3432 cm^{-1}) compared to pristine MMT, indicating the intercalation of CNFs into MMT nanofluidic membrane. What's more, the characteristic peak appeared in 848 cm^{-1} was assigned to the Al-O-C vibrations (*Science*, 2007, 318, 80-83), indicating the formation of covalent binding between MMT nanosheets and CNFs. Consequently, the chemical binding between MMT nanosheets and CNFs contributes to an interlocking configuration formed in the 2D-NNF, which is similar as the stable "brick and mud" configuration of natural nacre. Such a configuration enables a superior structure stability and high mechanical strength of the nanofluidic membrane. Compared with pristine MMT membrane, the 2D-NNF with an interlocking configuration demonstrates a high tensile strength up to 115 MPa, which is comparable with natural nacre (~130 MPa). The superior structure stability and high mechanical strength make it possible to construct robust large-area nanofluidic membrane towards real-world applications in

osmotic energy harvesting. For a better understanding, we have added above discussions and corresponding figures in the revised manuscript (**Page 4, Fig. S3 and S4**), in order to demonstrate the interactions between MMT nanosheets and CNFs and their roles in improving structural stability and functionality of the membrane.

Fig. R19 (a) Schematic of the covalent bonds formed between Si-OH or Al-OH at the edge of MMT and CNFs. XPS spectra of C1s of (a) 2D-NNF and (b) pristine MMT nanofluidics. High spectra of 2D-NNF and MMT for (d) Al2p and (e) Si2p region. The characteristic peak Si 2p originated from Si-O-Si/Si-O-Al, and the main peak of Al 2p was attributed to the Al-OH and Al-O-Si. The binding energy displacement of Al 2p and Si 2p compared with pristine MMT membrane, which was mainly attributed to the formation of covalent bonds between MMT and CNFs. (f) The survey XPS of 2D-NNF and MMT nanofluidics. For the 2D-NNF, C1s and O1s peak enhancement due to the introduction of CNFs. For pristine MMT nanofluidics, the appearance of carbon may be due to surface adsorption with weak strength and there is no obvious C=O peak in C1s region.

Fig. R20 FTIR spectrum of CNF, pristine MMT nanofluidics and 2D-NNF, the marked characteristic peaks correspond to the functional groups. The enhanced emerged peak at 3432 cm^{-1} and 1627 cm^{-1} corresponds to -OH and C=O group vibrations, respectively, which confirmed the successful introduction of CNFs. Besides, the decrease of free OH on the surface of 2D-NNF at 3621 cm^{-1} indicates the occurrence of dehydration condensation reaction.

6. What specific mechanisms facilitate the rapid transport of cations within the developed two-dimensional nanofluidic system, particularly considering the interplay between montmorillonite nanosheets and cellulose nanofibers?

Response: Thanks for the very helpful comment. We believe that the rapid transport of cations in the nanoconfined 2D channels follows classical hopping mechanism as the proton does in proton exchange membrane. The hopping mechanism was initially proposed to explain the ion transport behavior when the EDL is fully covered the whole nanochannels. In this work, the intercalation of negatively-charged CNFs into the interlayers of MMT nanosheets enlarged EDL region in the nanochannels to guarantee the applicability of this mechanism. In order to understand the hopping mechanism of cation diffusion process, the simulations of K^+ ion transport in negatively-charged 2D nanofluidic membrane were performed via Ab initio molecular dynamics (AIMD). Diffusion EDL was mainly derived from Gouy-Chapman-Stern model (*Phys. Fluids.*, 2005, 17, 100604), which consists of the Stern layer and diffuse layer. While, the diffuse

layer was composed of K^+ in our modeling two-dimension all-natural nanofluidic (2D-NNF)/ K^+ system. The 2D-NNF was coated with a layer of K^+ , and each K^+ has interactions with the O at the surface of 2D-NNF. At the beginning of MD simulations, the additional K^+ with velocity collided with one K^+ at rest in the diffuse layer, and the other K^+ in the diffuse layer moved with pendulum-like motion, along the additional K^+ velocity direction. K^+ far away from the collision moved 1.5 nm following 80 ps in the diffuse layer, through pendulum-like motion shown in **Fig. R21**. Thus, the rapid K^+ transport in developed 2D nanofluidics presented energy transfer hopping process according to the hopping mechanism. In order to improve the depth of the paper, we have added above discussions on the mechanism of rapid cation transport in our revised manuscript (**Page 7, Fig. S13**).

Fig. R21 Ab initio molecular dynamics simulations of rapid K^+ transport in diffuse layer according to hopping mechanism.

7. How were the surface and space negative charges effectively incorporated into the interlamellar channels, and what role did these charges play in enhancing the selective transport of cations, leading to the significant osmotic power output? It is supposed that the authors comment on these issues.

Response: Thank you for the helpful comments. The construction of 2D-NNF is based on the intercalation of negatively-charged CNFs into the interlamellar channels of negatively-charged MMT nanosheets. In this configuration (**Fig. R22**), the excess

negative charge on MMT wall is termed as surface charge and the negative charge from the intercalated CNFs is usually called space charge (*Nat. Commun.*, 2019, 10, 2920; *J. Am. Chem. Soc.*, 2021, 143, 1932; *Nano Energy*, 2022, 92, 106709). Both the surface charge and space charge could form EDLs for selective transport of cations as discussed above. For the pristine MMT nanochannels with only surface charge, the formed EDL region may not fully cover the whole channels (**Fig. R21, upper**), especially for the high-concentration solutions, which reduces the cation selectivity and thus affects the osmotic energy harvesting. When introducing the space charge of CNFs, the EDL region was enlarged and covered almost the whole nanochannels (**Fig. R22, bottom**), contributing to the enhanced ion selectivity and thus an improved osmotic energy harvesting. The above conclusions have been verified by the reported experimental results that the creation of space charge in the nanochannels contributes to significant osmotic power output from nanofluidic systems (*Nat. Commun.* 2019, 10, 2920; *J. Am. Chem. Soc.* 2022, 144, 13764-13772). For a better understanding, we have added above discussions in the revised manuscript (**Page 5, Fig. S8**).

Fig. R22 Schematic diagram of surface charge and space charge incorporated into the interlamellar nanochannels.

8. The authors should provide more details about the parameters and methodologies used in the life cycle assessment. What were the key findings regarding the economic, environmental, and energy benefits in comparison with other existing osmotic energy generation methods?

Response: Thank you for these helpful comments. Following the process of life cycle assessment, we selected 5 kg prepared materials as the functional unit, and determined the system boundary for assessment, as shown in **Fig. 6c** in the manuscript. We have added an input and output inventory on the processes of MMT, GO, MXene, CNF and ANF preparation, as shown in **Table R5**. We further utilized the inventory and applied the life cycle assessment method of CML2001 (2016 revised version) to obtain the characterization results of environmental impacts, such as GWP (global warming potential), AP (acidification potential), HTP (human toxicity potential), MAETP (marine aquatic eco-toxicity potential), FAETP (freshwater eco-toxicity potential), ADP (abiotic depletion) and so on. Meanwhile, we have added a detailed explanation of the TOPSIS method (**SI Page5-7**), which mainly used to calculate the weight proportion of multi-objective decision making in **Fig. 6d** in the manuscript. Furthermore, the price of industrial grade purity was estimated for raw materials and chemical reagents in **Table S7**, supplemented by the main reference survey website: (<https://www.100ppi.com/>, <https://www.mysteel.com/>, <https://p4psearch.1688.com/>). The prices of raw materials and chemical reagents in different periods will fluctuate. For **Table R6**, detailed production processes of 2D materials and nanocellulose used for nanofluidics preparations in reported works (*Nat. Commun.*, 2019, 10, 2920; *Energy Environ. Sci.*, 2021, 14, 4400; *Mater. Horiz.*, 2020, 7, 2702; *ACS Cent. Sci.*, 2021, 7, 1486-1492; *Nano Energy*, 2022, 100, 107526) were summarized to evaluate the resource, environment and economic impacts of the mainstream 2D material-based nanofluidics. We added above content in the revised manuscript (**SI Page 4, 5-7, 37**).

Table R5 Input and output inventory of each material preparation.

Category	Input and Output	Unit	Process					
			MMT Preparation	GO Preparation	MXene Preparation	CNF Preparation	ANF Preparation	
Energy	Electricity	MJ	1.08E+02	7.08E+02	1.11E+03	4.46E+02	9.77E+02	
	Deionised water	kg	8.60E+03	1.04E+04	5.70E+03	5.90E+03	2.00E+01	
	Bentonite clay	kg	3.00E+02					
	Sodium phosphate	kg	3.00E+00					
	Sodium hydroxide	kg		1.79E+01		2.02E+00	5.00E-01	
	Sulfuric acid	kg		6.57E+02			2.50E+01	
	Potassium permanganate	kg		5.22E+01				
	Hydrogen peroxide	kg		8.55E+02				
	Sodium nitrate	kg		7.15E+00				
	Hydrochloric acid	kg		1.17E+02	1.76E+02			
	Graphite	kg		1.79E+01	6.00E+00			
	Titanium	kg			9.00E+00			
	Resource	Aluminum	kg			3.60E+00		
		Lithium fluoride	kg			1.49E+01		
Hard wood		kg				1.20E+01		
Sodium bromide		kg				6.00E-01		
Sodium hypochlorite		kg				1.32E+00		
Sodium sulfide		kg				8.60E-01		
Calcium chloride		kg					2.40E+00	
Dimethylsulfid		kg					2.50E+02	
Benzal chloride		kg					2.00E+00	
N,N-dimethylformamide		kg					4.00E+01	
Terephthaloyldichloride		kg					4.00E+00	
Potassium hydroxide		kg					5.00E+00	
MMT		kg	5.00E+00					
GO		kg		5.00E+00				
Product	MXene	kg			5.00E+00			
	CNF	kg				5.00E+00		
	ANF	kg					5.00E+00	
Pollutant	Waste water	t	3.70E+00	6.50E+00	7.00E-01	8.00E-02	1.40E-01	

TOPSIS calculation method

According to the different decision-making objectives, the environment, resource, and economy factors are coupled through the method of multi-objective decision-making to ensure the objectivity of target weight setting. Among the multi-objective decision-making methods, the TOPSIS method is to establish the initial decision matrix and standardize it. By calculating the distance between each scheme relative to the optimal and worst scheme, the closeness between each scheme and the optimal scheme is obtained. Its dimensionless method for indicators can effectively solve the incommensurability between environment, resource, and economy factors, and the construction of positive and negative ideal schemes can effectively achieve the purpose of scheme sorting and selection.

(1) The multi-objective decision analysis model is constructed based on TOPSIS method

The basic principle of TOPSIS is to sort the existing schemes according to the proximity between the evaluation object and the idealized target to determine the optimal scheme. Among them, the positive and negative ideal solutions are assumed as the best and worst schemes. The positive ideal solution refers to an ideal solution that each index reaches the optimal; while the negative ideal solution is the worst case for each index value. The evaluation steps of TOPSIS multi-objective decision-making method are as follows:

Suppose that there are several schemes: $R = \{r_1, r_2, \dots, r_m\}$, each scheme has $\{f_1, f_2, \dots, f_n\}$ evaluation indexes, and the initial matrix is as follows:

$$R = (f_{ij})_{m \times n} = \begin{vmatrix} f_{11} & f_{12} & \cdots & f_{1n} \\ f_{21} & f_{22} & \cdots & f_{2n} \\ \dots & \dots & \dots & \dots \\ f_{m1} & f_{m2} & \cdots & f_{mn} \end{vmatrix}$$

where m and n are the number of solutions, the dimension of the evaluation index. Here $n=3$, refers to the three evaluation indexes of environment, resource, and economy, respectively.

① Standardized decision matrix: to eliminate the incommensurability between indicators.

$$r_{ij} = \frac{f_{ij}}{\sqrt{\sum_{i=1}^m f_{ij}^2}} \quad (i = 1, 2, \dots, m; j = 1, 2, \dots, n)$$

where r_{ij} is standardized decision matrix. The environment, resource, and economy factors can be compared objectively after the standardization process.

② Weighted standardized decision matrix : the decision indexes are weighted respectively to obtain the weighted standardized decision matrix.

$$V_{ij} = r_{ij} \times w_j \quad (i = 1, 2, \dots, m; j = 1, 2, \dots, n)$$

where V_{ij} refers to the weighted normalized decision matrix, w_j refers to the weight of each decision index f_j , $\sum_{j=1}^n w_j = 1$.

③ Calculate positive and negative ideal solutions.

$$A^+ = \{v_1^+, v_2^+, \dots, v_n^+\} = \{(max v_{ij} | j \in I'), (min v_{ij} | j \in I'')\}$$

$$A^- = \{v_1^-, v_2^-, \dots, v_n^-\} = \{(max v_{ij} | j \in I'), (min v_{ij} | j \in I'')\}$$

where A^+ and A^- are the positive and negative ideal solutions. For the revenue index, the maximum value of the row vector is a positive ideal solution, while the opposite is true for cost-based indexes. In this study, because the target scheme is to achieve the lowest resource consumption, environmental impact and the economic cost, the three evaluation indexes are all cost-based, that is, the minimum value of the row vector is a positive ideal solution.

④ Calculate the distance between the superior and inferior solutions, and sort the schemes:

$$d_i^+ = \sqrt{\sum_{j=1}^n (v_{ij} - v_j^+)^2} \quad (i = 1, 2, \dots, m)$$

$$d_i^- = \sqrt{\sum_{j=1}^n (v_{ij} - v_j^-)^2} \quad (i = 1, 2, \dots, m)$$

where d_i^+ refers to the distance between the evaluation scheme and the positive ideal solution, d_i^- refers to the distance between the evaluation scheme and the negative ideal solution.

According to the r_i^* score from high to low, the final ranking of the schemes can be considered as the best scheme.

$$r_i = \frac{d_i^-}{d_i^+ + d_i^-} (i = 1, 2, \dots, m)$$

(2) Model sensitivity analysis

Based on the above TOPSIS method, the sensitivity analysis is further integrated to calculate the change level of the final results of environment, resource, and economy factors under different weight distribution modes, and to investigate the stability of the evaluation results. When the weights of environment, resource, and economy factors change from 0 to 1, the ranking of different schemes is recalculated to further determine the stability interval of the evaluation results (**Fig. 6d in the manuscript**).

For the key findings regarding the economic, environmental, and energy benefits in comparison with other existing osmotic energy generation methods. We have added three methods other for the use of mainstream 2D nanofluidics for osmotic energy generation in **Table R6**, in order to fully demonstrate the unique advantages of 2D-NNF. By comparing the data of economic benefits, resource consumption and environmental emissions, the key findings are as follows:

For the economic benefits, the cost in the production process of 2D-NNF is only 1/4 and 1/13 of GO and MXene nanofiber-based membrane, respectively, which can be attributed to the abundant reserves of natural bentonite, and primarily utilizes physical methods for substance purification and without the use of strong corrosive and oxidizing reagents. From an economic standpoint, the complex preparation process of GO and the high cost of MXene raw materials, specifically metal powder, further hindered their practical application.

For the environmental emissions, compared with GO fiber-based nanofluidics, 2D-NNF reduce the emissions of greenhouse gases (i.e., CO₂, CH₄) and acidic gases (i.e., SO₂) by about an order of magnitude during the production process. This is mainly ascribed to the fact that the production process almost does not involve the use of corrosive chemical reagents, thus it rarely causes subsequent environmental pollution.

Furthermore, the human toxicity potential (HTP) during the production of 2D-NNF is reduced by two orders of magnitude compared to the use of LiF and HCl during the exfoliation of MXene. For resource consumption, compared with MXene fiber-based nanofluidics, the production process of 2D-NNF does not involve the consumption of non-renewable resource such as titanium and aluminium, as well as high-energy consumption such as ball milling and calcination. Therefore, its energy consumption and abiotic depletion potential (ADP fossil) level in the production process are only 2/15 and 1/14 of GO and MXene nanofiber-based membrane, respectively. For a better understanding, we have added above discussions in the revised manuscript (**Page 16, SI Page 39**).

Furthermore, some values of the original Table S8 has been corrected, and the follow supplementary **Table R6** was added to the revised version. (**SI Page 39, Table S10**)

Table R6 Comparisons of this work (2D-NNF) with the reported 2D nanofluidics from the resource, environment and economy benefits.

Nanofluidic Membrane	2D-NNF	GO/ANF	MXene/CNF	GO/CNF	MXene/ANF	MMT/ANF
GWP (kg CO ₂ e)	8.27E+01	7.60E+02	4.79E+02	6.70E+02	5.46E+02	1.53E+02
AP (kg SO ₂ e)	3.72E-01	4.81E+00	1.51E+00	4.77E+00	1.65E+00	5.22E-01
HTP (kg DCBe)	3.01E+00	4.51E+01	1.83E+02	4.35E+01	1.85E+02	5.20E+00
MAETP (kg DCBe)	3.03E+03	3.33E+04	1.35E+05	3.40E+04	1.34E+05	4.66E+03
FAETP (kg DCBe)	2.44E-01	3.35E+00	1.81E+00	3.49E+00	2.02E+00	4.70E-01
POCP (kg Ethenee)	2.40E-02	2.89E-01	1.71E-01	2.73E-01	1.82E-01	3.6E-02
EP (kg Phosphatee)	5.12E-02	1.70E-01	1.28E-01	1.54E-01	1.43E-01	6.71E-02
Energy consumption (kWh)	4.24E+01	2.51E+02	3.19E+02	2.79E+02	3.37E+02	6.26E+01
ADP fossil (MJ)	1.14E+03	1.59E+04	6.56E+03	1.53E+04	8.03E+03	2.76E+03
Corrosive reagent consumption (kg)	3.10E+00	9.94E+02	1.91E+02	9.40E+02	2.22E+02	3.74E+01
Cost (USD)	1.05E+02	3.84E+02	1.26E+03	3.83E+02	1.25E+03	1.30E+02

9. What strategies were employed to ensure the long-term stability of the large-area membrane over the course of 30 days? What potential challenges and solutions were encountered during this period?

Response: Thank you for this helpful comment. During the long-term measurement, concentration diffusion between the two solutions across the membrane will cause the decrease in the salinity gradient, which in turn reduce the osmotic energy harvesting. Thus, a periodical electrolyte replenishment was conducted to achieve a fixed salinity gradient for the long-term stability of membrane over 30 days. However, a human interference to the electrolyte of osmotic energy harvesting system will inevitably cause fluctuation of the measured output current and thus affect the accuracy of experimental results. For long-term experimental study, it is a big challenge to maintain fixed salinity gradient across the membrane like natural junction of sea and river. To solve this problem, the peristaltic pumps could be used to maintain constant concentrations of solutions across the membrane via a continuous renewal of the solutions. This is considered as an effective solution for achieving stable and continuous osmotic energy output for a long-term study (*J. Am. Chem. Soc.*, 2022, 144, 12400-12409).

10. In practical terms, how versatile is this technology across different environmental conditions? Are there specific environmental factors (temperature, salinity, etc.) that might influence its efficiency and stability in real-world osmotic energy harvesting applications?

Response: Thank you for the helpful comments. In order to evaluate the universality of the 2D-NNF developed in this work, we have studied the influences of environmental temperature and salinity on the osmotic energy harvesting. As shown in **Fig. R23**, the output current, power density and energy conversion efficiency of 2D-NNF gradually increase with the temperature. This phenomenon is mainly attributed to that a high temperature will reduce the liquid viscosity and increase the ionic mobility (*Nanoscale*, 2017, 9, 12068; *Phys. Chem. Chem. Phys.*, 2022, 24, 20303). Furthermore, the average output power could maintain at about 14.43 W m^{-2} for a long-term measurement. The

results shows that a higher temperature is beneficial to achieve high and stable osmotic energy output.

Fig. R23 (a) Current density and (b) power density output at different temperature. (c) Maximum power generation and energy conversion efficiency under different temperature (concentration gradient is 0.5 M/0.01 M NaCl). (d) Lifetime of power output stability under the load resistance of ~ 6 kΩ at 313 K. Error bars represent S.D.

Furthermore, various salinities that simulate different water environments were constructed to study the performance of 2D-NNF under high polarization conditions. The artificial seawater was replaced by artificial brackish water (0.2 M), desalination brine (1.3 M), mining waste water (3.0 M) and salt-lake water (4.3 M). As shown in **Fig. R24**, the output current, power density of 2D-NNF gradually increase with the salinity gradient, while the energy conversion efficiency decreases slightly given the ionic concentration polarization. A maximum power density of 60.46 W m^{-2} could be achieved from the 2D-NNF based osmotic energy harvesting system at the salt lake/river condition. Furthermore, the diffusion current could maintain a continuous operation for a long-term measurement without electrolyte replenishing. The influence

of these environment factors could be considered in the development of high-performance nanofluidics towards the real-world osmotic energy harvesting applications. For a better understanding, we have added above discussions in our revised manuscript (Page 11, 13, Fig. S24 and S27).

Fig. R24 (a) The current densities and (b) power densities at different salinity conditions as a function of the external resistance. Power generation and conversion efficiency (c) and long-term current output (d) under a series of artificial water resources including brackish water, desalination brines, brines from mining activities, and water from salt-lake. Error bars represent S.D.

11. Please edit the language carefully, fix typos, and correct grammatical errors.

Response: Thanks for the helpful comments. We have checked the manuscript carefully and corrected some language, fix typos, and grammatical errors (Page 2, 3, 4, 6, 8, 10, 11, 12, 20). For a better understanding, some poor statements have also been optimized. For instance:

(1) *Original description:* For the large-scale membrane with an area of 700 cm^2 , the average maximum power still reaches 8.36 W m^{-2} and could maintain long-term

stability over 30 days, attributing to excellent uniformity and stability in both physical and chemical structures. **(Line 99 of the original manuscript)**

Revised description: For the large-scale membrane with an area of 700 cm², the osmotic energy conversion was carried out by selecting several test sites from the different regions of membrane under the same test conditions. The average maximum power of different test sites still reaches 8.36 W m⁻² and could maintain long-term stability over 30 days, attributing to excellent uniformity and stability in both physical and chemical structures. **(Line 102-107, page 3 in the revised manuscript)**

(2) *Original description:* The velocity ratio (V_{K^+}/V_{Cl^-}) between K⁺ and Cl⁻ ions in the vertical direction was 10 times larger than V_{K^+}/V_{Cl^-} in horizontal direction. **(Line 234-236 in original manuscript)**

Revised description: The velocity ratio (V_{K^+}/V_{Cl^-}) between K⁺ and Cl⁻ ions in the horizontal direction was 1/10 of that V_{K^+}/V_{Cl^-} in vertical direction. **(Line 306-308, page 9 in revised manuscript)**

(3) *Original description:* When the same salinity gradient was applied to KCl solution, the power density was up to 2.82, 10.38 and 17.06 W m⁻², respectively (Fig. S20), which mainly attributed to different ionic diffusion coefficients ($K^+ > Na^+$). **(Line 293-295 in original manuscript)**

Revised description: For a fair comparison, the same salinity gradients were applied to the KCl aqueous solution system, that is the high-salinity solution was fixed at 0.5 M KCl and the low-salinity one was controlled as 1, 10 and 100 mM. In this case, the obtained osmotic power density was respectively 2.82 (5-fold), 10.38 (50-fold) and 17.06 W m⁻² (500-fold) as shown in Fig. S25. A higher osmotic energy output in KCl system compared with NaCl system was mainly attributed to the higher ionic diffusion coefficient of K⁺ ion ($1.960 \times 10^{-9} \text{ m}^2 \text{ s}^{-1}$) than Na⁺ ion ($1.334 \times 10^{-9} \text{ m}^2 \text{ s}^{-1}$). **(Line 372-378, page 11 in revised manuscript)**

(4) *Original description:* Regarding nanocellulose, the environmental impact of CNF was found to be 10 times lower than that of ANF at an equivalent mass (Table S6), despite the involvement of a small amount of chemical reagents in the preparation process. **(Line 417-420 in original manuscript)**

Revised description: Regarding the nanocellulose, the environmental impact of CNF was 1/10 of that of ANF at an equivalent mass (Table S8), despite the involvement of a small amount of chemical reagents in the preparation process. **(Line 503-505, page 15 in revised manuscript)**

(5) *Original description:* Consequently, the energy consumption and abiotic depletion (ADP) in the production process of 2D-NNF is reduced by a factor of 10.2 and 13.9, respectively. Additionally, the corresponding reductions in GWP, AP, HTP, MATEP, and FAETP are 9.2, 12.9, 60.9, 44.5, and 14.0 times, respectively, in comparison to the GO and MXene nanofiber-based membrane (Table S8). **(Line 440-444 in original manuscript)**

Revised description: Consequently, the energy consumption and abiotic depletion (ADP fossil) in the production process of 2D-NNF are only 2/15 and 1/139 of GO and MXene nanofiber-based membrane consumption, respectively. Additionally, the corresponding impact in GWP, AP, HTP, MATEP, and FAETP are 1/9, 1/13, 1/61, 1/45 and 1/14, respectively (Table S10). **(Line 522-526, page 16 in revised manuscript)**

We hope the revised manuscript could meet the standards of this journal.

REVIEWERS' COMMENTS

Reviewer #2 (Remarks to the Author):

The authors answered all my questions.

Reviewer #3 (Remarks to the Author):

My comments have been properly addressed. I agree with the publication of this manuscript in the current format.